# Sensitivity of Northern Hemisphere climate to ice-ocean interface heat flux parameterizations

Xiaoxu Shi[1], Dirk Notz[2,3], Jiping Liu[4], Hu Yang[1], and Gerrit Lohmann[1]

[1]Alfred Wegener Institute, Helmholtz center for Polar and Marine Research, Bremerhaven, Germany
[2]Institute for Oceanography, Center for Earth System Research and Sustainability (CEN), Hamburg University, Hamburg, Germany
[3]Max Planck Institute for Meteorology, Hamburg, Germany
[4]Department of Atmospheric and Environmental Sciences, University at Albany, New York, USA

**Correspondence:** Xiaoxu Shi (xshi@awi.de)

**Abstract.** We investigate the impact of three different parameterizations of ice-ocean heat exchange on modeled sea-ice thickness, sea-ice concentration, and water masses. These three parameterizations are (1) an ice-bath assumption with the ocean temperature fixed at the freezing temperature; (2) a 2-equation turbulent heat-flux parameterization with ice-ocean heat exchange depending linearly on the temperature difference between the underlying ocean and the ice-ocean interface whose temperature is kept at the freezing point of the sea water; and (3) a 3-equation turbulent heat-flux approach in which the ice-ocean heat flux depends on the temperature difference between the underlying ocean and the ice-ocean interface whose temperature is calculated based on the local salinity set by the ice-ablation rate. Based on model simulations with the standalone sea-ice model CICE, the ice-ocean model MPIOM and the climate model COSMOS, we find that compared to the most complex parameterization (3), the approaches (1) and (2) result in thinner Arctic sea ice, cooler water beneath high-concentration ice and warmer water towards the ice edge, and a lower salinity in the Arctic Ocean mixed layer. In particular, parameterisation (1) results in the smallest sea ice thickness among the 3 parameterizations, as in this parameterisation all potential heat in the underlying ocean is used for the melting of the sea ice above. For the same reason, the upper ocean layer of the central Arctic is cooler when using parameterisation (1) compared to (2) and (3). Finally, in the fully coupled climate model COSMOS, parameterisations (1) and (2) result in a fairly similar oceanic or atmospheric circulation. In contrast, the most realistic parameterization (3) leads to an enhanced Atlantic meridional overturning circulation (AMOC), a more positive North Atlantic Oscillation (NAO) mode and a weakened Aleutian Low.

## 1 Introduction

The growth and decay of sea ice at the ice–ocean interface are determined by the local imbalance between the conductive heat flux within the ice and the oceanic heat flux from below the ice. Because the temperature at the ice–ocean interface is

determined by phase equilibrium, any imbalance between the two fluxes is not compensated by changes in the local temperature as is the case at the ice surface, but instead by ice growth or ablation. This makes the evolution of sea ice thickness very sensitive to small changes in oceanic heat flux (e.g. Maykut and Untersteiner, 1971). Thus a realistic parameterization of flux exchanges at the ice-ocean interface is important for simulating sea ice and its climate feedback.

For the simplest parameterization, the ice–ocean system is simply treated as an ice bath (c.f. Schmidt et al., 2004): The temperature of the uppermost ocean grid cells is fixed at the freezing point temperature, and any excess energy that enters these grid cells via advection, convection, or heat exchange with the atmosphere is instantaneously applied to the ice through lateral and bottom melting. Such parameterization is consistent with turbulence models that treat the flux of heat and salt as analogous to momentum flux (Josberger, 1983; Mellor et al., 1986), which results in very efficient transfer whenever the ice is in motion

relative to the underlying water. However, the ice-bath paradigm is incompatible with observations, i.e., the 1984 Marginal Ice Zone Experiment (MIZEX), which clearly indicates that an ice-covered mixed layer can store significant amounts of heat (i.e., remain above freezing) for extended periods of time (McPhee, 1986; McPhee et al., 1987; Perovich and Maykut, 1990). These measurements demonstrate that in particular during melting, the exchange of scalar quantities such as heat and salt differs significantly from the exchange of momentum. The reason for this lies in the fact that, unlike ice-ocean momentum flux, heat

and mass transfer are strongly affected by a thin sublayer controlled by molecular processes (McPhee et al., 1987). Consistent with laboratory studies of heat transfer over hydraulically rough surfaces (Yaglom and Kader, 1974), the heat exchange is hence not only determined by turbulent processes but also by diffusion through this molecular sublayer.

The fact that the oceanic temperature can be significantly higher than the freezing temperature even underneath a dense ice cover cannot be represented in numerical models that employ an ice-bath assumption. More advanced formulations of

ice–ocean heat exchange are therefore based on bulk formula, where the ice–ocean heat exchange depends linearly on the temperature difference between the mixed layer and the ice–ocean interface. Early models used a constant diffusion term as the proportionality constant (Røed, 1984), while more advanced formulations made the heat exchange depend on friction velocity as well (McPhee, 1992). For such more advanced formulations, measurements show proportionality between heat flux and temperature difference times friction velocity across a large range of Reynolds numbers (e.g., Fig. 6.5 in McPhee, 2008).

These formulations form the basis of many modern sea-ice models (e.g. Hunke and Lipscomb, 2010).

These formulations, despite being physically much more realistic than the crude ice-bath assumption, often suffer from the fact that the temperature at the ice–ocean interface is simply set as the freezing temperature of the underlying sea water. In reality, however, the interfacial temperature is determined by a *local* phase equilibrium, and in particular during periods of high ablation rates, the local salinity at the interface can be significantly lower than the salinity of the sea water underneath. The

interfacial temperature can be significantly higher than the freezing temperature of the underlying sea water. Therefore, one extension of the turbulent parameterizations of ice–ocean heat exchange lies in the explicit calculation of the temperature at the ice–ocean interface based on local salinity (Jenkins et al., 2001; Notz et al., 2003; Schmidt et al., 2004). Such formulations then allow for the explicit calculation of heat and salt fluxes, and give a more realistic estimate of ice–ocean heat exchange. In particular, these formulations allow for the ice–ocean interface to be warmer than the underlying sea water, which allows for

heat fluxes from the interface into the underlying ocean. This becomes particularly important when large amounts of meltwater

accumulate underneath sea ice during summer (Notz et al., 2003; Tsamados et al., 2015). For such more advanced formulations, measurements show proportionality between heat flux and temperature difference times friction velocity across a large range of Reynolds numbers (e.g., Fig. 6.5 in McPhee, 2008). These formulations form the basis of many modern sea-ice models (e.g. Hunke and Lipscomb, 2010).

Exploring the behavior of different parameterizations describing ice-ocean heat flux has been an important topic in model studies. Significant differences can generally exist between melt rates calculated with the 3-equation approach and less realistic approaches (c.f. Notz et al., 2003; Tsamados et al., 2015), as only the 3-equation approach allows for heat fluxes that are directed from the interface into the water and therefore allows for a realistic limitation of melt rates through the formation of a fresh water layer underneath the ice. A previous study has examined the sensitivity of sea ice simulation to the approaches

introduced in McPhee (1992) and Notz et al. (2003) using a stand-alone sea ice model CICE (Tsamados et al., 2015). CICE uses a simple thermodynamic slab ocean with fixed mixed layer depth and sea water salinity. Thus, the realistic effect of oceanic processes can not be represented. For example the sea ice over the Southern Ocean is severely overestimated by CICE due to a lack of warming effect from the Antarctic deep water. Therefore it is necessary to also investigate the ice-ocean heat flux formulations in a more complex system, including an interactive ocean or even the atmosphere. Based on this motivation, in the

present study, we examine how different physical realism that is represented by the three discussed parameterizations impact the resulting ice cover, large-scale oceanic circulation, and atmosphere properties in different numerical models ranging from an idealized columnar model, a stand-alone sea ice model, an ice-ocean coupled model, to a most complex climate system model.

     Another motivation of our study is to help improve the formulation describing ice-ocean heat flux in various models. For

example in the 4th version of CICE, only ice-bath and 2-equation assumptions could be applied. In MPIOM and COSMOS, the ice-bath approach is used, which can lead to an overestimation of oceanic heat flux into sea ice. In our study we implemented the more realistic 3-equation parameterization into all the models mentioned above.

     The paper is organized as follows. Section 2 describes the parameterizations in details. Section 3 introduces various models that we use for our purposes: 1) A conceptual 1-dimensional model allows us to examine a wide parameter range. 2) The Los

Alamos Sea Ice Model (CICE) allows us to determine changes of sea ice in a modern sea-ice model. 3) The Max-Planck-Institute Global Ocean/Sea-Ice Model (MPIOM) can be used for examining the impact of the parameterizations on the large-scale ocean circulation. 4) The fully coupled climate model COSMOS can further help us to look at the atmospheric response to the described parameterizations. Section 4 gives an overview of our experiment configuration. Section 5 describes results from sensitivity studies with using the various models. We discuss and summarize our main findings in section 6.

## 2    Heat flux parameterizations

The growth and decay rate $\dot{h}$ of sea ice at the ice–ocean interface is determined by the imbalance of the conductive heat flux into the ice and the oceanic heat flux $F_{oce}$ from underneath the ice. Hence,

$$\rho_i L \dot{h}(t) = k_i \frac{\partial T}{\partial z}\Big|_{ice} + F_{oce}, \tag{1}$$

where $\rho_i$ is the density of the ice, $L$ is the latent heat of fusion, $k_i$ is thermal conductivity of the ice, $T$ is temperature and $z$ is the vertical coordinate. Some simple sea-ice models assume that sea ice has no heat capacity and does not absorb solar radiation. In these so-called zero-layer models, the temperature gradient is constant and simply given as the temperature difference between the ice surface and the ice bottom, divided by ice thickness (Semtner Jr, 1976). In more advanced sea-ice models, the ice consists of several layers and the conductive heat flux into the lower most grid cell is explicitly calculated.

As discussed in the introduction, a number of approaches exist for the calculation of the oceanic heat flux $F_{oce}$ (c.f. Holland and Jenkins, 1999; Jenkins et al., 2001; Notz et al., 2003; Schmidt et al., 2004). For the simplest parameterization, the ice–ocean system is simply treated as an ice bath: The temperature of the uppermost ocean grid cells is fixed at its freezing temperature, and any excess energy that enters these grid cells via advection, convection, or heat exchange with the atmosphere is instantaneously applied to the ice through lateral and bottom melting. Hence, the oceanic heat flux is given as

$$F_{oce} = \frac{\rho_w c_w (T_{mix} - T_f) h_{mix}}{\delta t}, \tag{2}$$

where $T_{mix}$ is ocean temperature, $T_f$ is the salinity-dependent freezing temperature, $\rho_w$ is the density of the sea water and $c_w$ the specific heat capacity, all determined for the uppermost oceanic grid cell with vertical extent $h_{mix}$. $\delta t$ is the time step.

In more realistic formulations, the heat flux is determined from a bulk equation based on friction velocity and temperature difference between the mixed layer and the ice–ocean interface according to

$$F_{oce} = -\rho_w c_w \alpha_h u_* (T_{mix} - T_{interface}), \tag{3}$$

where $u_*$ is friction velocity and $\alpha_h$ is a turbulent heat exchange coefficient (McPhee et al., 2008). A number of different formulations exist for the calculation of the interfacial temperature. Following Schmidt et al. (2004), these can be differentiated between a 1-equation approach, a 2-equation approach and, most realistically, a 3-equation approach. In the 1-equation approach, $T_{interface}$ is simply set to a constant value. We will not consider this approach any further here. In the more realistic 2-equation approach, $T_{interface}$ is set to the freezing temperature of the sea water in the upper-most ocean grid cell. Hence, in addition to Eq. (3), the freezing-point relationship of seawater is also required, which is the second equation of the 2-equation approach:

$$T_{interface} = -0.054 \cdot S_{interface} \tag{4}$$

In this most realistic 3-equation approach, $T_{interface}$ is set to the freezing temperature of the water that exists directly at the interface. The salinity of this water is explicitly calculated from a salinity-balance equation:

$$(S_{interface} - S_{ice})\dot{h}(t) = \alpha_s u_* (S_{mix} - S_{interface}). \tag{5}$$

Here, $S_{interface}$ is the salinity directly at the interface, which decreases during melting through the addition of fresher melt water of sea ice with salinity $S_{ice}$. Salt is exchanged with the underlying water (with salinity $S_{mix}$) through turbulent exchange,

with a salt exchange coefficient $\alpha_S$. Together with the freezing equation 1 of sea water, Equations (3) and (5) form the three equations of the 3-equation approach. These three equations can be solved to calculate the three unknowns $\dot{h}$, $S_{interface}$ and

$T_{interface}$.

As mentioned before, only the 3-equation approach allows for heat fluxes that are directed from the interface into the water. In addition, only the 3-equation approach allows for a realistic limitation of melt rates through the formation of a fresh water layer underneath the ice. For these reasons, significant differences can generally exist between melt rates calculated with the 3-equation approach and less realistic approaches (c.f. Notz et al., 2003).

Quantitatively, a value of $0.005 < \alpha_h < 0.006$ has been found to give good agreement between measured and calculated heat fluxes for a large spread of Rayleigh numbers (McPhee, 2008; McPhee et al., 2008). More uncertainty exists regarding the most appropriate values for the turbulent exchange coefficients $\alpha_h$ and $\alpha_s$ for the 3-equation approach. Their ratio $R = \alpha_h/\alpha_s$ depends on the molecular diffusivities for heat and salt as well as on the roughness of the boundary (McPhee et al., 1987; McPhee, 2008). Laboratory experiments imply $35 \leq R \leq 70$ (Owen and Thomson, 1963; Yaglom and Kader, 1974; Notz

et al., 2003). Sirevaag (2009) found from an analysis of field data, $R \approx 33$, while McPhee et al. (2008) suggest a value of $R \approx 35$. During freezing conditions, salt and heat are transported almost equally efficiently (McPhee et al., 2008). This is because during freezing conditions, the water column is statically unstable owing to the salt release from growing sea ice. Hence, during freezing conditions, $R \approx 1$ (McPhee et al., 2008), and the 2-equation approach can be used without much loss in accuracy. Best agreement with observational data is then found for $\alpha_h = 0.0057$.

In testing the impact of the various parameterizations on modeled sea ice and ocean circulation, we therefore take the following approach: for the ice-bath parameterization, we simply incorporate Eq. (2). For the 2-equation approach, we use Eq. (3) with $\alpha_h = 0.006$ and the freezing-point relationship for seawater. For the 3-equation approach, we differentiate between freezing and melting conditions. During melting, we use the full 3-equation approach with $R = 35$ as our reference. In an idealized 1-D model used in our study, $R = 70$ is also applied to test the sensitivity with respect to this parameter. For a certain

value of $R$, we calculate $\alpha_h$ to satisfy the requirement described in McPhee et al. (1999). $R = 35$ is associated with a turbulent heat exchange coefficient of $\alpha_h = 0.0095$; and $R = 70$ with $\alpha_h = 0.0135$. During freezing, we fall back to the 2-equation approach.

## 3  Models

We will now briefly introduce the four different models that we use to analyse the different response to oceanic heat-flux

parameterizations based on the ice-bath assumption, the 2-equation approach and the 3-equation approach. We start with a description of our idealized columnar model with simple sea ice thermodynamics, then move to the stand-alone sea ice model CICE, and finally describe the ice–ocean model MPIOM and the fully coupled ice–ocean–atmosphere model COSMOS.

## 3.1 Idealized 1-D model

We use a one-dimensional columnar sea-ice model coupled to a simple ocean mixed layer to carry out sensitivity studies and to investigate the impact of the three formulations for ice–ocean heat exchange in an idealised setup.

The model consists of a simple zero-layer sea-ice model, where the surface temperature $T_s$ is determined by balancing atmospheric fluxes and the conductive heat flux through the ice according to

$$-(1-\alpha)F_{sw} - F_{other} + \epsilon\sigma T_s{}^4 = -k_i\frac{T_s - T_{bot}}{h}. \tag{6}$$

Here, $\alpha$ is the albedo of the ice surface, $F_{sw}$ is the short-wave flux, $\epsilon = 0.95$ the infrared emissivity, $\sigma = 5.67 \times 10^{-8}$ the Stefan-Boltzman constant, $T_{bot}$ the temperature at the ice-ocean interface, and $F_{other}$ is the sum of sensible heat flux, latent heat flux and downward longwave radiation flux. $(1-\alpha)F_{sw} + F_{other}$ represents the heat input to the surface of the ice, and $\epsilon\sigma T_s{}^4$ the upward longwave radiation flux from the ice surface. For simplicity, we assume that the thermal conductivity of sea ice $k_i$ is constant and set $k_i = 2.03$ W/(m· K) according to the 1-D thermodynamic sea-ice model of Maykut and Untersteiner (1971). During melting periods, the surface temperature is fixed at the bulk freezing temperature of the ice and the excess heat is used to melt ice at the surface. We assume the sea ice in our idealized model to be very fresh, using a freezing temperature of 0 °C. At the ice bottom, the model calculates the change in ice thickness by balancing the conductive heat flux and the oceanic heat flux according to Eq. (1).

The seasonal variation of the atmospheric fluxes $F_{sw}$ and $F_{other}$ are prescribed according to the fits provided by Notz (2005), approximating the monthly-mean observational data compiled by Maykut and Untersteiner (1971). These fits are:

$$F_{sw} = A_1 \exp[B(\frac{d - C_1}{D_1})^2] \tag{7}$$

$$F_{other} = A_2 \exp[B(\frac{d - C_2}{D_2})^2] + E \tag{8}$$

where $A_1$=314, $A_2$=117.8, B=-0.5, $C_1$=164.1, $C_2$=206, $D_1$=47.9, $D_2$=53.1, $E$=179.1, and $d$ is the number of the day in the year. The seasonal variation in surface albedo is calculated as

$$\alpha = \frac{F}{1 + (\frac{d-G}{H})^2} + I \tag{9}$$

where F, G, H and I have the values of -0.431, 207, 44.5 and 0.914, respectively. This equation is a fit to measurements obtained during the Surface Heat Budget of the Arctic Ocean (SHEBA) campaign (Perovich et al., 1999).

The model is coupled to an idealised oceanic mixed layer of depth $h_{mix}$, which can store and release heat. The coupling between the mixed layer ocean and the sea ice via the oceanic heat flux $F_{oce}$ is given by the three parameterizations as described before.

## 3.2 CICE

To investigate the sensitivity of sea ice to the three ice-ocean heat flux parameterizations in a modern sea-ice model, we use version 4.0 of the stand-alone sea ice model CICE. The model consists of a multi-layer energy-conserving thermodynamic sub-model (Bitz and Lipscomb, 1999) with a sub-grid scale ice-thickness distribution, and a submodel of ice dynamics based on an elastic-viscous-plastic rheology (Hunke and Dukowicz, 1997; Hunke, 2001) that uses incremental remapping for ice advection (Lipscomb and Hunke, 2004). A detailed model description is given in Hunke and Lipscomb (2010).

The surface temperature of the ice is calculated by balancing incoming fluxes from the atmosphere with outgoing longwave fluxes and the conductive heat flux in the ice. For the albedo, we here use the standard setup of The Community Climate System Model version 3 (CCSM3), where the (spectral) albedo is calculated explicitly based on snow and ice temperature and thickness (see Hunke and Lipscomb (2010) for details). A bulk sea ice salinity of 4 psu is implemented.

We run CICE in standalone mode, coupled to the mixed-layer ocean that forms part of the CICE package. The heat flux between this mixed-layer ocean and the sea ice is in the standard form of CICE described by the 2-equation approach with $\alpha_h = 0.006$. This formulation is here either used directly or replaced by the ice-bath formulation or the 3-equation formulation as described before. The salinity of the mixed-layer in SIM and CICE is kept at 34 g/kg.

## 3.3 MPIOM

To examine the interaction of changes in the sea-ice model with large-scale ocean circulation, we use the ocean general circulation model MPIOM (Max-Planck Institute Ocean Model). MPIOM is based on the primitive equations with representation of thermodynamic processes (Marsland et al., 2003). The orthogonal curvilinear grid is applied in MPIOM with the north pole located over Greenland. The relevant terms of the surface heat balance are parameterized according to bulk formulae for turbulent fluxes (Oberhuber, 1993), and radiant fluxes (Berliand, 1952).

The sea-ice component of MPIOM uses zero-layer thermodynamics following Semtner Jr (1976) and viscous-plastic dynamics following Hibler III (1979). It does not allow for a sub-grid ice-thickness distribution. The sea-ice state within a certain grid cell is hence fully described by ice concentration $C$ and ice thickness $h$. The surface heat balance is solved separately for the ice covered and ice free part of every grid cell. Any ice that is formed through heat loss from the ice-free part is merged with the existing ice to form a new ice thickness and ice concentration. The change in sea ice thickness and concentration can be calculated via two main ice distribution parameters as outlined by Notz et al. (2013), the first one being a so-called lead closing parameter that describes how quickly the sea ice concentration increases during new ice formation processes, and the other describing the change in the ice-thickness distribution during melting. In its standard setup, MPIOM uses an ice-bath parameterization to calculate the heat flux between the ocean and the ice. Wherever covered by sea ice, the temperature of sea water in the uppermost grid cell is kept at its freezing point. All heat entering the uppermost grid cells either from the atmosphere or from the deeper oceanic grid cell is instantaneously transported into the sea-ice cover, maintaining the temperature of the uppermost oceanic layer at the freezing point. In the present study, this formulation is either used directly, or replaced by the 2-equation or the 3-equation parameterization.

 ## 3.4 COSMOS

The comprehensive climate model COSMOS (ECHAM5-MPIOM), developed by the Max Planck Institute for Meteorology, is used in the present study to further investigate the atmospheric response to the three ice-ocean heat flux parameterizations. The atmosphere component ECHAM5 solves the primitive equations for the general circulation of the atmosphere on a sphere (Roeckner et al., 2003). It is formulated on a Gaussian grid for the horizontal transport schemes and on a hybrid sigma-pressure grid for the vertical coordinate. The OASIS3 coupler (Valcke, 2013) is used for the coupling between the ocean and the atmosphere components. Once per simulated day, solar and non-solar heat fluxes, hydrological variables, and horizontal wind stress are provided from the atmosphere to the ocean through OASIS3. At the same time, the ocean provides its sea-ice coverage and the sea-surface temperature to the atmosphere.

## 4 Experimental design

For each model, we perform separate simulations based on the three ice-ocean heat flux formulations (see table 1). We assume that the 3-equation approach with $R = 35$ describes reality more realistically, and hence use this simulation as our reference. In our idealized 1-D model, we also use $R = 70$ to test the model sensitivity with respect to this parameter. For a given value of $R$, we calculate $\alpha_h$ to satisfy the requirement described in McPhee et al. (1999). This results in a turbulent heat exchange coefficient $\alpha_h = 0.0095$ for $R = 35$ and $\alpha_h = 0.0135$ for $R = 70$. In SIM and CICE the mixed-layer salinity has a constant value of 34 g/kg. In MPIOM and COSMOS the salinity of the sea water evolves dynamically in response to oceanic or atmospheric processes.

As atmospheric forcing, we use for our conceptual 1-D model equations (7) to (9), For CICE, we use the National Center for Atmospheric Research (NCAR) monthly-mean climatological data with $1° \times 1°$ resolution Kalnay et al. (1996). Input fields contain monthly climatological sea-surface temperature, sea-surface salinity, the depth of ocean mixed layer, surface wind speeds, 10 m air temperature, humidity and radiation for the time period of 1984-2007. For MPIOM, we use a GR30 (about $3°$) horizontal resolution and 40 uneven vertical layers, forced by daily heat, freshwater and momentum fluxes as given by the climatological Ocean Model Intercomparison Project (OMIP) forcing (Röske, 2006).

For the coupled model COSMOS, the configuration of the ice-ocean component MPIOM is the same as we use for the stand-alone version of this model. The atmospheric module ECHAM5 is used at T31 resolution ($3.75°$) with 19 vertical levels. The coupled model was initialized from a pre-industrial simulation and integrated with the solar constant, Earth's orbital parameters and greenhouse gas forcing all fixed at their 1950 CE values. All simulations were run sufficiently long to reach quasi-equilibrium.

## 5 Results

We now turn to a description of the simulated responses of sea ice, ocean and atmosphere to the three different parameterizations.

**Table 1.** List of experiments, note that $R = \alpha_h/\alpha_s$, denoting the ratio between turbulent exchange coefficients for heat ($\alpha_h$) and salt ($\alpha_s$).

| Name | Parameterization | $T_{interface}$ | $T_f$ | length (model year) |
|---|---|---|---|---|
| SIM-icebath | Ice bath | same as $T_f$ | $-1.84\,^{\circ}C$ | 100 |
| SIM-2eq | 2-equation | same as $T_f$ | $-1.84\,^{\circ}C$ | 100 |
| SIM-3eq35 | 3-equation, with R=35 | from Eq. (1), Eq. (3), Eq. (5) | $-1.84\,^{\circ}C$ | 100 |
| SIM-3eq70 | 3-equation, with R=70 | from Eq. (1), Eq. (3), Eq. (5) | $-1.84\,^{\circ}C$ | 100 |
| CICE-icebath | Ice bath | same as $T_f$ | $-1.84\,^{\circ}C$ | 100 |
| CICE-2eq | 2-equation | same as $T_f$ | $-1.84\,^{\circ}C$ | 100 |
| CICE-3eq35 | 3-equation, with R=35 | from Eq. (1), Eq. (3), Eq. (5) | $-1.84\,^{\circ}C$ | 100 |
| MPIOM-icebath | Ice bath | same as $T_f$ | freezing point of the up-permost cell | 1000 |
| MPIOM-2eq | 2-equation | same as $T_f$ | freezing point of the up-permost cell | 1000 |
| MPIOM-3eq35 | 3-equation, with R=35 | from Eq. (1), Eq. (3), Eq. (5) | freezing point of the up-permost cell | 1000 |
| COSMOS-icebath | Ice bath | same as $T_f$ | freezing point of the up-permost cell | 1000 |
| COSMOS-2eq | 2-equation | same as $T_f$ | freezing point of the up-permost cell | 1000 |
| COSMOS-3eq35 | 3-equation, with R=35 | from Eq. (1), Eq. (3), Eq. (5) | freezing point of the up-permost cell | 1000 |

## 5.1 Influence of $u_*$, $h_{mix}$ and ice concentration

We start with a number of sensitivity experiments with our simple 1-D model that were carried out to understand the underlying relationship between simulated ice thickness and the three different parameterizations. We performed four different simulations with our simple model, which in the following are called SIM-icebath, SIM-2eq, SIM-3eq35 and SIM-3eq70, where for the 3-equation setup the last number denotes the value of $R = \alpha_h/\alpha_s$. We run the simulations until the ice reaches its equilibrium thickness, with no more changes from one year to the next.

In our standard SIM simulations we applied $u_* = 0.002\ m/s$, a sea-ice concentration $C = 85\%$, an albedo of sea water $\alpha_{oce} = 0.1$ and a mixed-layer depth $h_{mix} = 40$ m. The sea ice salinity is kept at 0. In winter, when the ocean loses energy to the atmosphere through the open-water part of the grid cell, the simulated heat loss from the ocean is identical in the four setups (Fig. 1c), since their open-water part is identical and the ocean is constantly at its freezing temperature (Fig. 1b). Hence, any

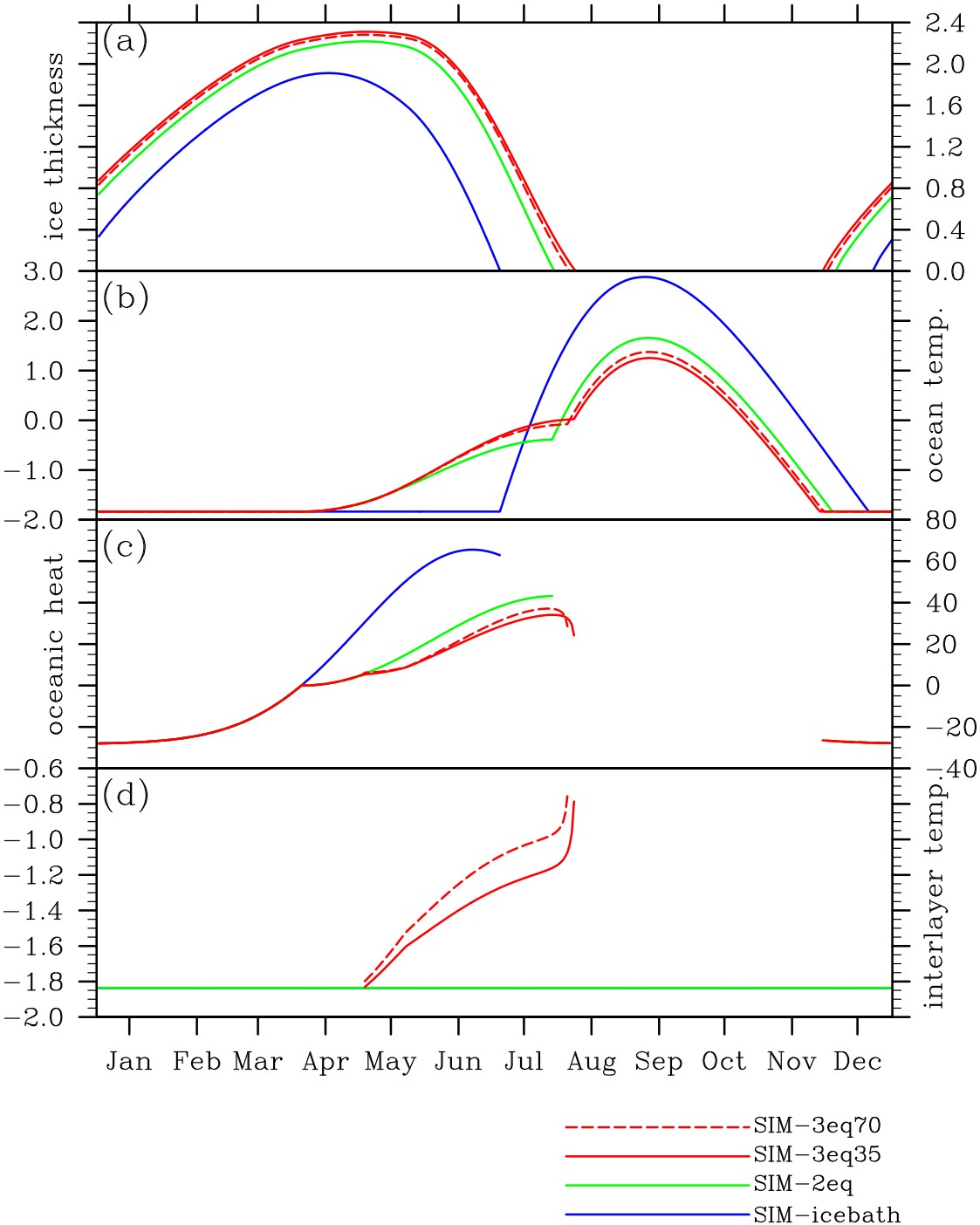

**Figure 1.** Time series of (a) sea-ice thickness (units: m), (b) ocean temperature (units: $^{\circ}C$), (c) ocean-to-ice heat flux (units: W/m$^2$) and (d) ice-ocean interface temperature (units: $^{\circ}C$) in the experiments SIM-icebath, SIM-2eq, SIM-3eq35 and SIM-3eq70 with friction velocity at 0.002 and ice concentration at 75%. The model is run into equilibrium.

heat that is extracted from the mixed layer directly causes ice growth, which explains the very similar accretion rates of the sea ice (Fig. 1a). Major differences between the simulations arise as soon as the net heat flux becomes positive and begins to heat the ocean. All energy that enters the ocean is then directly used to melt the ice in SIM-icebath, while some of the heat is stored in the ocean in SIM-2eq and SIM-3eq. Hence, ice in SIM-2eq and SIM-3eq melts slower than the ice in SIM-icebath, and the

ocean remains warmer throughout spring (Fig. 1b). Once the ice in SIM-icebath is melted completely, the ocean temperature rises rapidly and quickly exceeds that in SIM-2eq and SIM-3eq. This can be explained by two facts: 1) in SIM-2eq and SIM-3eq, the sea ice reflects most of the incoming shortwave radiation, and 2) in SIM-2eq and SIM-3eq the heat flux absorbed by open water is primarily used for sea-ice melting, while in SIM-icebath no more sea ice exists such that the entire heat flux into the ocean causes a warming of the sea water. The slower melting of the ice in SIM-2eq and SIM-3eq and the resulting lower

heat storage in the ocean throughout summer results in an earlier onset of sea-ice formation during autumn.

For SIM-icebath and SIM-2eq, the temperature at the ice-ocean interface is constant at the freezing point of the sea-water, which for our choice of $S_{seawater} = 34$ g/kg is around $-1.84\,°C$. For SIM-3eq, the interface temperature can be significantly above this value, as the interface freshens through the melting of the comparably fresh sea ice (Fig. 1d).

Comparing SIM-2eq, SIM-3eq35 and SIM-3eq70, we find that the ice thins earlier and faster in SIM-2eq, because the

ocean heat flux between the ocean and the ice is amplified in this setup owing to the constantly cold interfacial temperature. Accordingly, the oceanic temperature increases more slowly throughout spring in SIM-2eq. In SIM-3eq70, the transport of salt to the interface is lower than in SIM-3eq35. Hence, the interface remains fresher and warmer throughout summer. Despite the warmer interface, stronger heat fluxes and slightly faster ablation of the ice are simulated, mainly resulting from a higher turbulent heat exchange coefficient $\alpha_h$, which is 0.0095 in SIM-3eq35 and 0.0135 in SIM-3eq70.

To quantify the different response of the simulated sea-ice cover for a larger range of forcing conditions, we carried out a series of sensitivity studies. For each of these, we varied one of the forcing parameters and analysed the difference in annual mean ice-ocean heat flux between SIM-3eq35 and SIM-icebath.

We find that in our simplified setup, differences in ice thickness between SIM-3eq35 and SIM-icebath increase with mixed layer depth. This is due to the fact that the same amount of heat input causes a smaller temperature change for a deeper mixed

layer. According to equation (3), a smaller temperature change then leads to a smaller change in heat flux to the ice bottom (Fig. 2).

In addition, we find that the difference in sea-ice thickness generally decreases with friction velocity. This is related to the fact that for larger friction velocity, more heat is transported to the ice-ocean interface in the 3-equation setup (Fig. 2), which enhances sea-ice melt.

Finally, regarding sea-ice concentration, we find in our simplified setup that differences in ice thickness between SIM-3eq35 and SIM-icebath are larger for a smaller ice concentration. This is related to the fact that the residual energy, which mainly comes from the net incoming heat flux through open water, is all used for ablating sea ice in SIM-icebath, while in SIM-3eq35 only a fraction of the heat is used for ice ablation. Lower ice concentration enhances the energy by the open water and therefore also the difference in the amount of heat transferred to the ice cover.

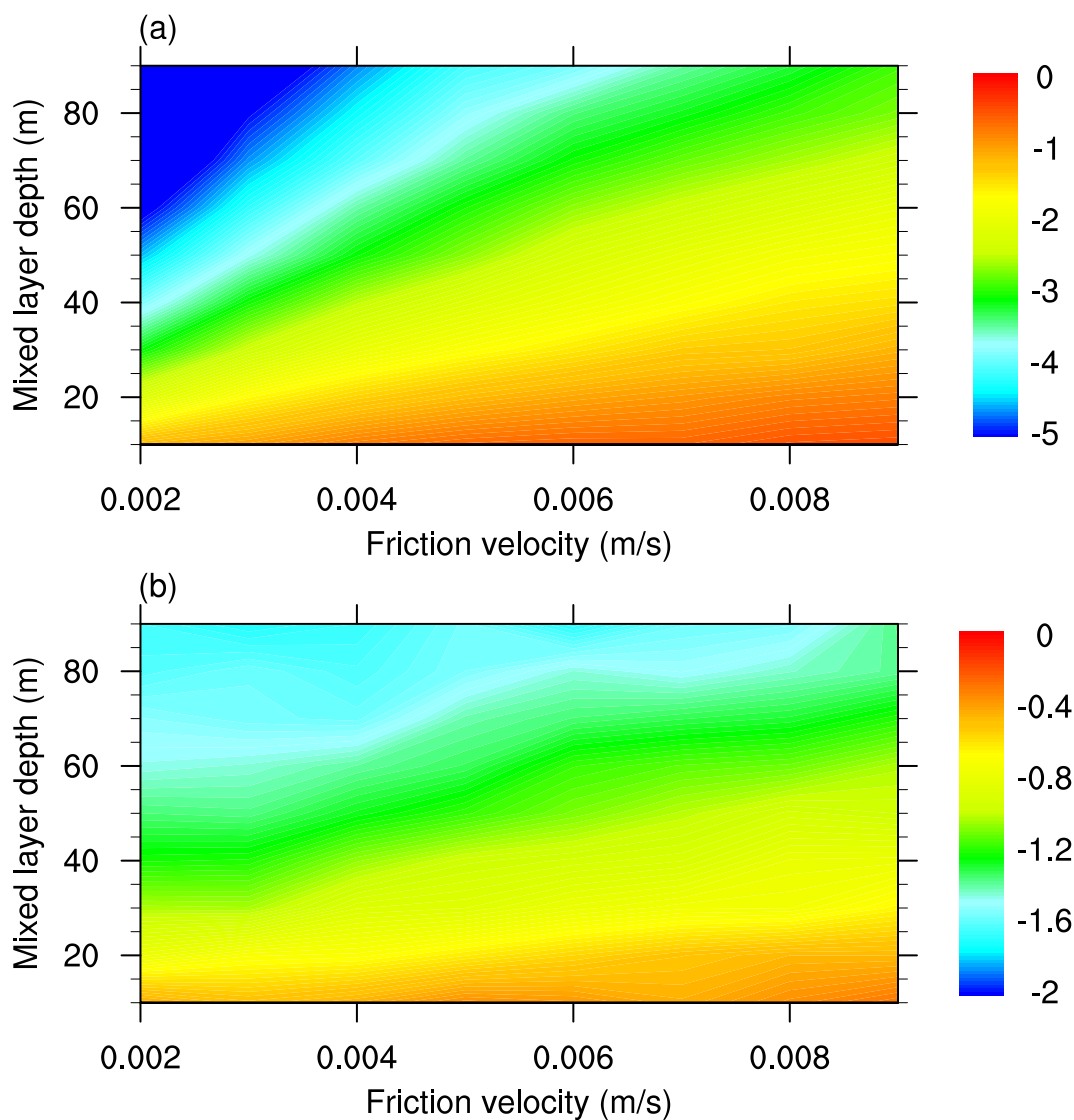

**Figure 2.** Anomalies of ocean-to-ice heat flux in SIM for (a) 3-equation minus icebath, and (b) 3-equation minus 2-equation, for different choices of mixed layer depth and friction velocity. Units are W/m$^2$.

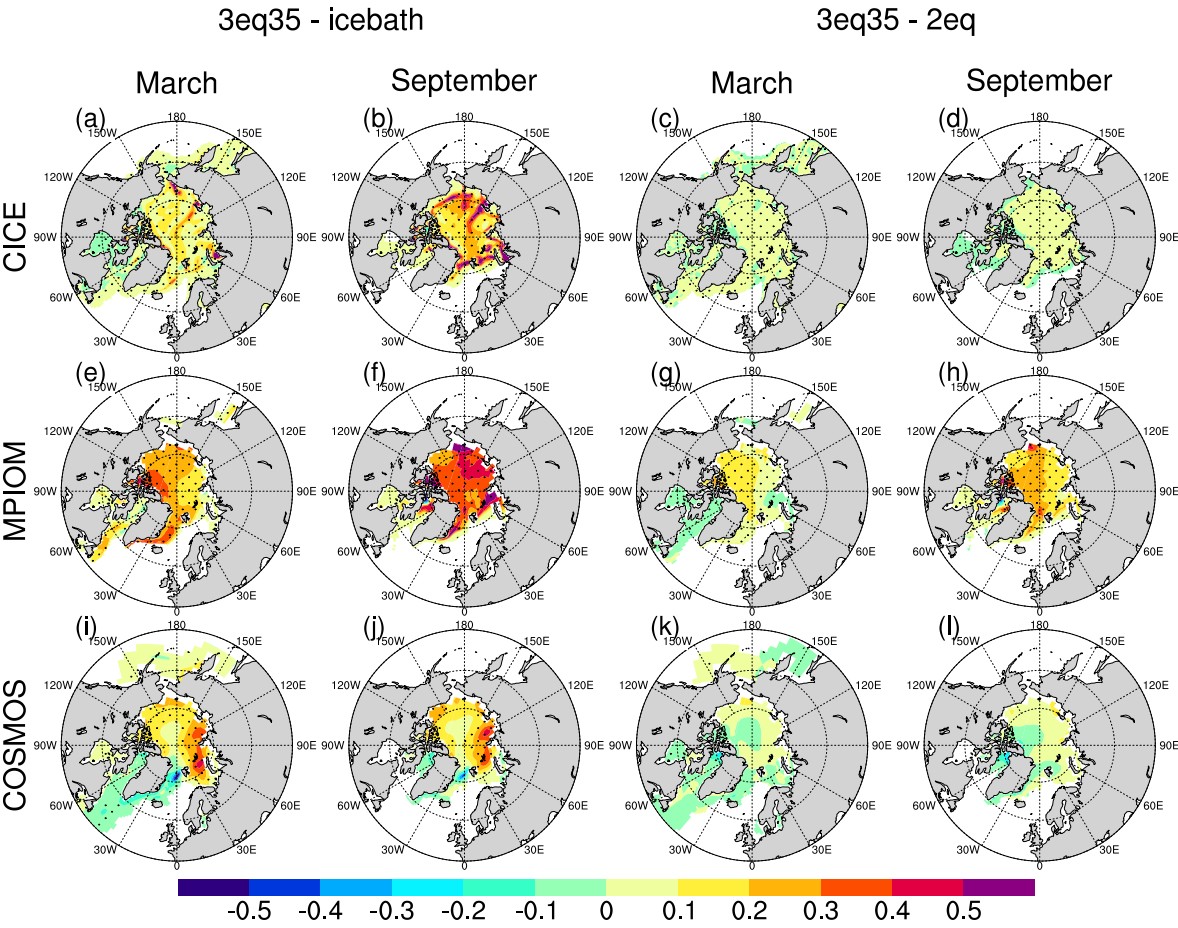

**Figure 3.** The difference in the Arctic sea-ice thickness for (a) CICE-3eq35 – CICE-icebath in March, (b) CICE-3eq35 – CICE-icebath in September, (c) CICE-3eq35 – CICE-2eq in March, (d) CICE-3eq35 – CICE-2eq in September, (e) MPIOM-3eq35 – MPIOM-icebath in March, (f) MPIOM-3eq35 – MPIOM-icebath in September, (g) MPIOM-3eq35 – MPIOM-2eq in March, (h) MPIOM-3eq35 – MPIOM-2eq in September, (i) COSMOS-3eq35 – COSMOS-icebath in March, (j) COSMOS-3eq35 – COSMOS-icebath in September, (k) COSMOS-3eq35 – COSMOS-2eq in March, and (l) COSMOS-3eq35 – COSMOS-2eq in September. The marked area has a significance level of greater than 95% based on Student's t-test. Units: m.

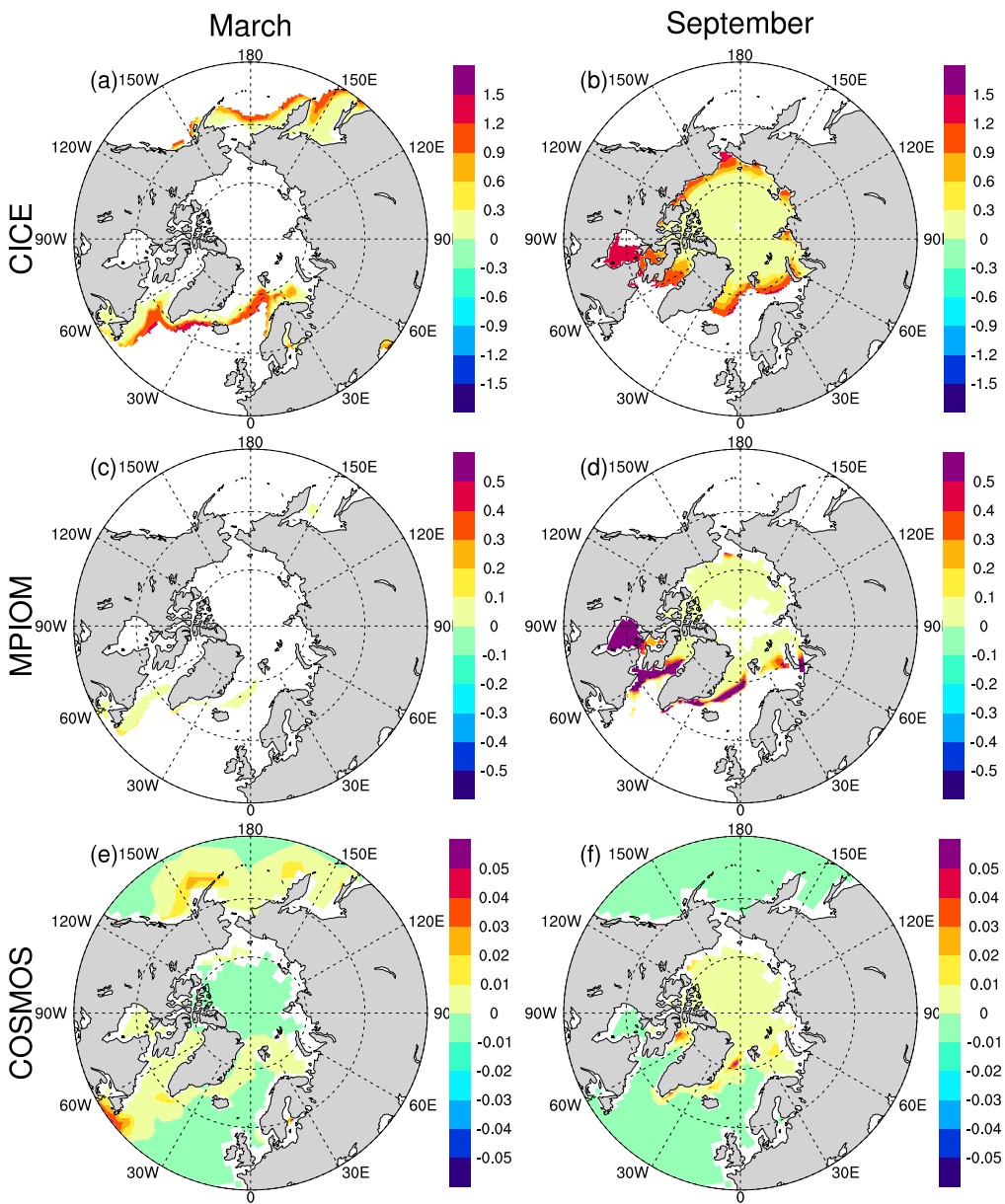

**Figure 4.** The anomaly of the Arctic ice-ocean interface temperature in (a-b) CICE-3eq35, (c-d) MPIOM-3eq35, and (e-f) COSMOS-3eq35 relative to freezing point of the far-field ocean (about $-1.8°$). The left column is for March and the right column for September. Units: K.

## 5.2 Ice thickness

Having understood some of the qualitative impact of the different parameterizations, we can now turn to an analysis of their impact in the more realistic setting provided by CICE, MPIOM and COSMOS. In these models $R = 35$ is applied in the full 3-equation approach. The presented results focus on the Arctic Ocean, as we only find a small response of Southern Ocean properties to the change of ice-ocean heat flux parameterizations in particular in MPIOM and COSMOS; furthermore, the stand-alone sea-ice model CICE simulates an unrealistic distribution of sea ice in warm months in the Southern Ocean, as it fails to capture the heat release from the relatively deep mixed layer.

We let all models run until the modeled ice cover and, in MPIOM and COSMOS, the deep ocean temperatures reach quasi-equilibrium. More concretely, we performed CICE experiments for 100 model years, with the last 10 years representing its quasi-equilibrium state. For MPIOM and COSMOS, 1000-model-year experiments were conducted, and data from the last 100 model years were used for analysis. The significance level of any differences between the individual simulations was calculated by performing Student's t-test which is used to examine if results from two different parameterizations are significantly different. For the Student's t-test, the interannual variances of the last 100 simulation years (10 years in the case of CICE) are considered.

We find that the ice thickness responds similarly to the different parameterizations as in the simple, one-dimensional model: Everything else unchanged, compared to the 3-equation approach the ice-bath parameterization leads to thinner ice throughout the Arctic Ocean, both in winter and summer (Fig. 3). The change is similar but less pronounced in the simulations based on the 2-equation parameterization. The most significant changes occur in the marginal ice zone where sea-ice concentration is lowest, again consistent with the results from the one-dimensional model. In the Arctic, the change in March thickness is generally less pronounced than the change in September thickness. This is due to the fact that the air-to-ocean heat flux tends to be negative (the ocean loses heat to the air) in March, and both the temperature of the water and the temperature at the ice-ocean interface are maintained at the freezing point. Hence, in all parameterizations, the extracted heat is directly transfered into sea-ice formation. In September, in contrast, the ocean can maintain a temperature above the freezing temperature in the 2-equation or 3-equation approach but not in the ice-bath approach. Hence, as in the simple 1-D model, differences between the different parameterizations are more pronounced during summer.

In addition, sea-ice concentration is high throughout March, which reduces the direct interaction of atmospheric heat fluxes with the ocean. As discussed in the previous subsection, this limits differences between the different parameterizations during winter time. Finally, the ice thins somewhat less in winter because of dynamical effects: the thinner ice is more mobile and more prone to ridging, which fosters the formation of areas with open water. In these areas, significant amounts of new ice can form, which dampens some of the thermodynamic thinning of the ice pack.

In summer, among the three parameterizations only the 3-equation approach can result in an ice–ocean-interface temperature above the freezing point of the uppermost ocean layer (Fig. 4), which reduces the ocean-to-ice heat flux. This is due to the fact that the ice-ocean interface is usually very fresh owing to the ablation of the ice bottom. When the temperature of the interface exceeds that of the mixed layer, a reversed heat flux from the ice to the ocean can occur.

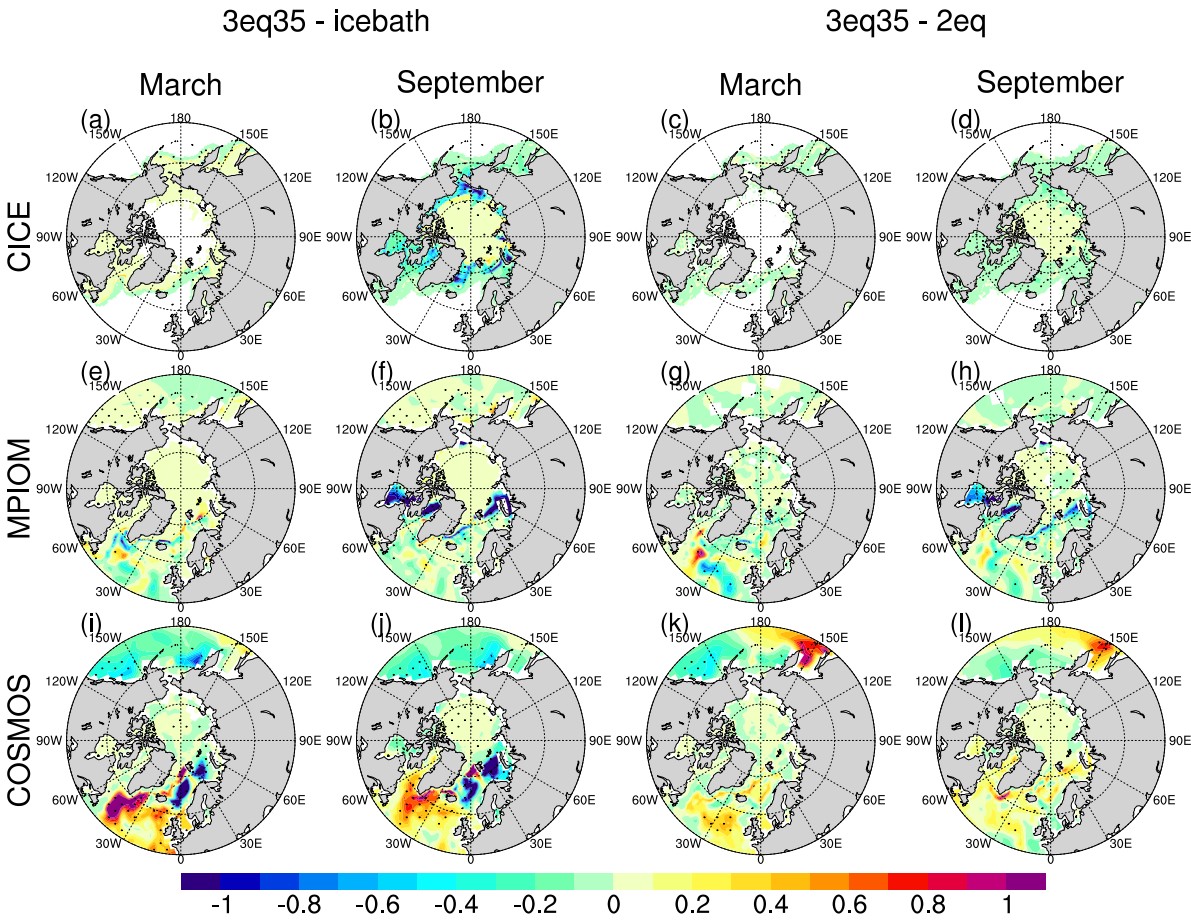

**Figure 5.** As in Fig. 3, but for the sea surface temperature. Units: K.

### 5.3 Upper ocean temperature and salinity

We now move on to analyse how the described changes in sea ice impact upper ocean temperature and salinity. We find for the Arctic Ocean that the ice-bath parameterization and the 2-equation approach result in almost the same temperature distribution during winter as the more realistic 3-equation approach in CICE and MPIOM (Fig. 5a,c,e,g). During summer, however, the ice-bath approach causes warmer water to persist around the ice edge in CICE (Fig. 5b). This is caused by the fact that here the ice melts earlier than in the 3-equation approach, which then allows the ocean to absorb heat more efficiently. The same is found in

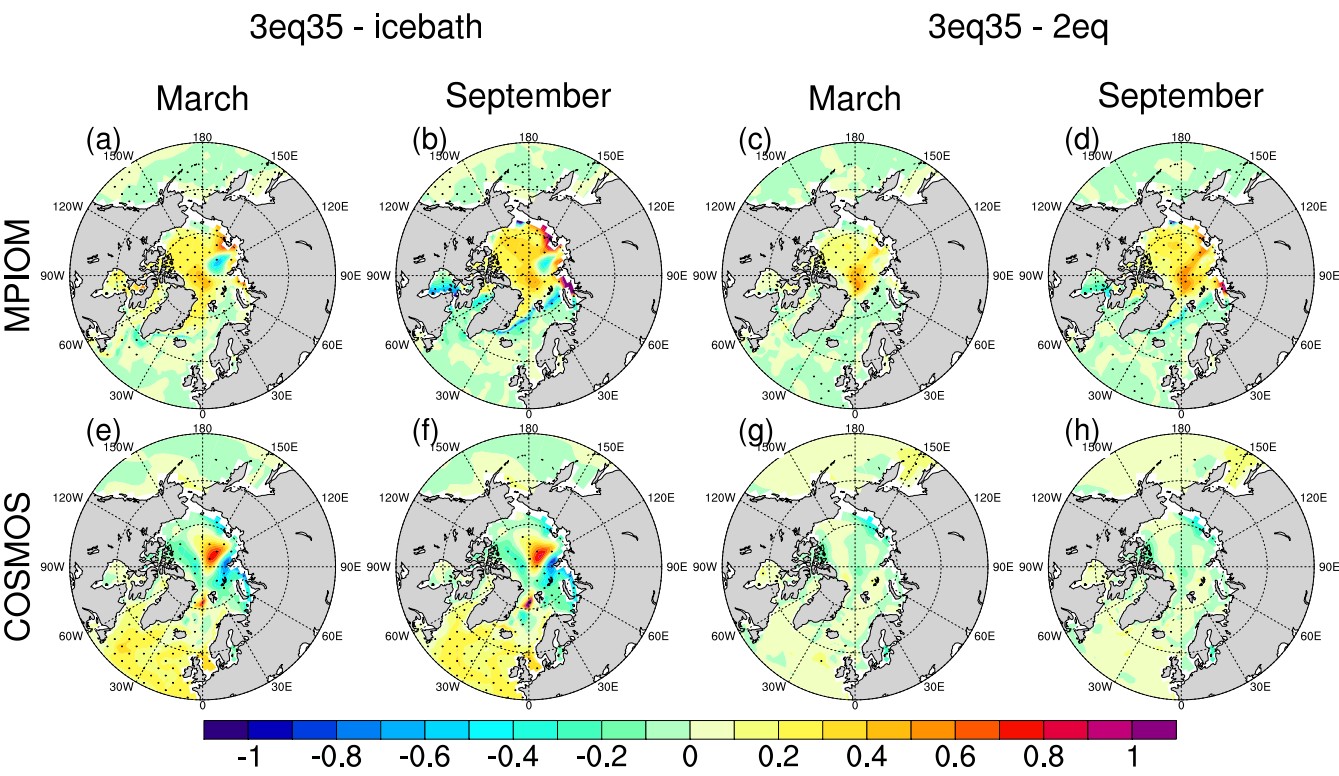

**Figure 6.** As in Fig. 3e-l, but for the sea surface salinity. Units: g/kg.

MPIOM in the areas of Hudson Bay, Baffin-Bay and Norwegian Sea and Barents Sea (Fig. 5f,h). The most intriguing feature
found in COSMOS is a significant cooling across the North Atlantic Ocean in the ice-bath and 2-equation parameterizations
compared to the 3-equation approach (Fig. 5i-l). Such cooling is a consequence of weakened thermohaline circulation which
tends to bring relatively warmer water from the lower latitudes (see Section 4.4).

Because brine is released from sea ice during its formation and growth, the changes in ice thickness between different
parameterizations should trigger changes in upper ocean salinity. Indeed, we find such changes to occur (Fig. 6): In regions in

which the ice-bath approach or the 2-equation approach cause an increased heat flux to the ice underside, and hence a larger melting rate of sea ice in summer as well as a smaller growth rate in winter, the ocean is generally less salty in the simulations with a simplified parameterization of ice–ocean heat exchange than in the simulations with the full 3-equation parameterization. Interestingly, the opposite sign is observed in the Barents sea and its adjacent regions (Fig. 6e-f), despite the larger melt rates in the ice-bath scheme. The North Atlantic Ocean experiences a pronounced freshening in the ice-bath approach in COSMOS (Fig. 6e-f), which lowers the efficiency of deep-water formation. No significant differences in upper ocean salinity are found between experiments COSMOS-2eq and COSMOS-3eq35 (Fig. 6g-h).

## 5.4   Thermohaline structure of the ocean

We now turn to the large-scale changes in the thermohaline structure of the ocean. We find that compared to the more realistic 3-equation approach, the ice-bath and 2-equation approaches lead to significant cooling of the ocean's deep water masses (Fig. 7c,e,g). This behaviour is due to the fact that the heat flux out of the ocean is slowed down in the 3-equation approach. Hence, more heat can be stored in the mixed layer and further advected into the deep ocean. However, the opposite is found in experiment MPIOM-icebath, which results in a pronounced warming in the deep water masses by up to 0.5 °C (Fig. 7a). This warming in the simulations with the least realistic parameterization of ice–ocean heat exchange reflects the earlier ice loss in the marginal ice zone, which causes enhanced surface warming of the water there.

As the simplified parameterizations both lead to faster melting of sea ice in the Arctic Ocean in summer, and less growth in winter, as compared to the most realistic approach, one would expect a freshening of the ocean mixed layer and the deep water mass that originates from such fresher surface source water. However, we find that such freshening in MPIOM occurs only within the Arctic upper ocean between depths of 0 and 100 m (Fig. 7b,d). In COSMOS, the freshening extends to the bottom of the Arctic Ocean (Fig. 7f,h). This different model behaviour is currently not understood.

The Atlantic meridional overturning circulation (AMOC) streamfunction, defined as the zonally integrated transport over the Atlantic basin, shows a weakening over 40-60°N, 0-3000 m depth in MPIOM-icebath and MPIOM-2eq compared to MPIOM-3eq35. In COSMOS, a pronounced weakening of AMOC is obtained south of 60°N. The AMOC index, i.e. the maximum value of the AMOC streamfunction over the region of 800-2000 m depth, 20-90°N is found to be 20.2 Sv and 17.6 Sv in MPIOM-3eq35 and COSMOS-3eq35, respectively (Table 2). The latter is consistent with the estimates of global circulation from hydrographic data (15±3 Sv) (Ganachaud and Wunsch, 2000). Compared to the corresponding 3-equation approach, the strength of the AMOC decreases by 1 Sv and 0.8 Sv in COSMOS-icebath and COSMOS-2eq respectively (Table 2). In COSMOS-icebath, the reduced sea surface salinity in the Atlantic section (Fig. 6e-f, Fig. 7f) lowers the efficiency of deep-water formation, resulting in a weakening of the AMOC (Fig. 8c). A similar but less pronounced pattern is obtained by COSMOS-2eq (Fig. 7h). No significant anomaly in the AMOC index is found in MPIOM (Table 2).

## 5.5   Atmospheric responses

We now finally turn to investigate how the sea ice changes affect the atmospheric properties in the fully coupled model COS-MOS.

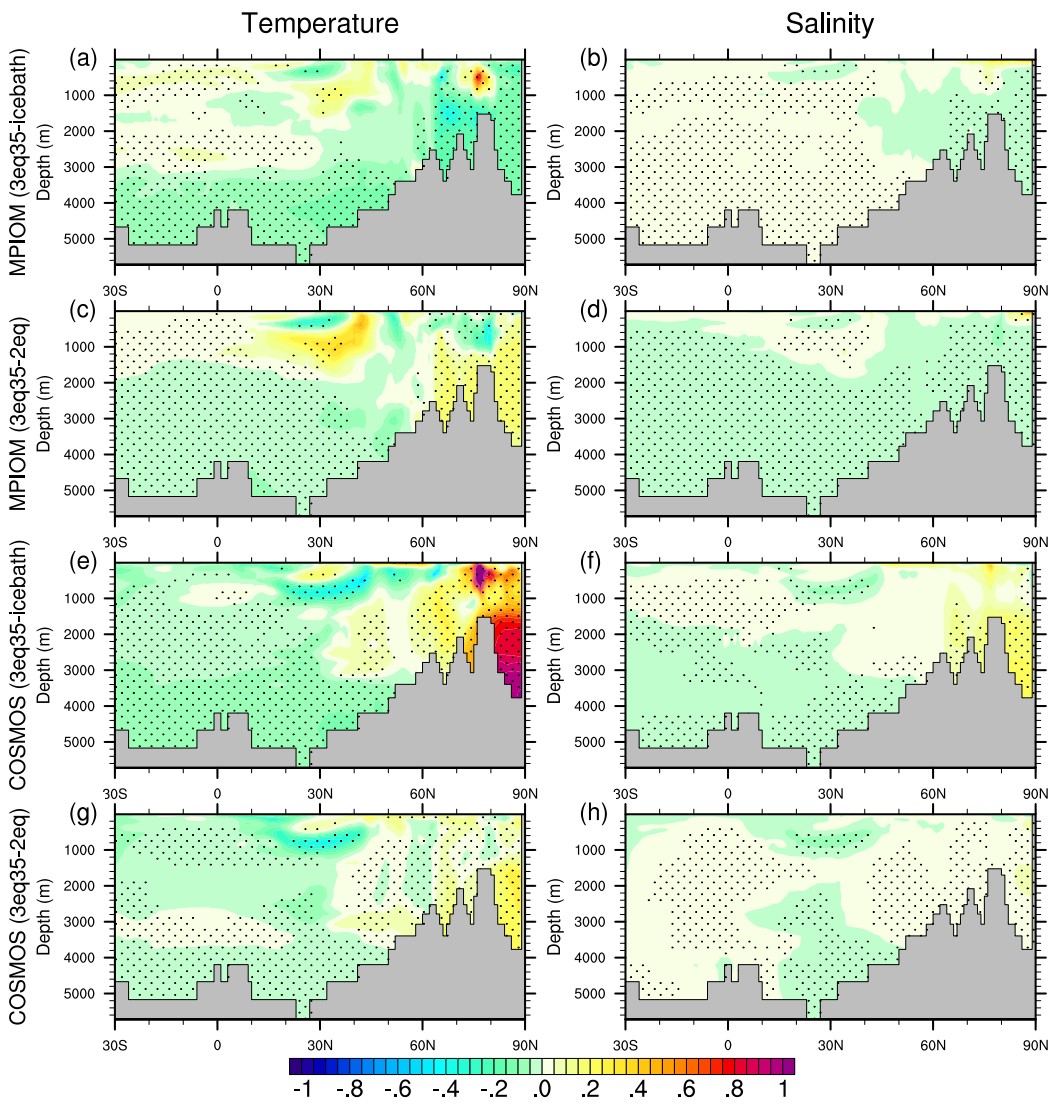

**Figure 7.** Anomalies in zonal mean temperature and salinity vertical profile across the North Atlantic section (-80-0°W) for the latitudes from 30°S to 90°N (a,b) MPIOM-3eq35 – MPIOM-icebath, (c,d) MPI-3eq35 – MPI-2eq, (e,f) COSMOS-3eq35 – COSMOS-icebath, and (g,h) COSMOS-3eq35 – COSMOS-2eq. The left column is for temperature and the right column for salinity. Units: K and g/kg.

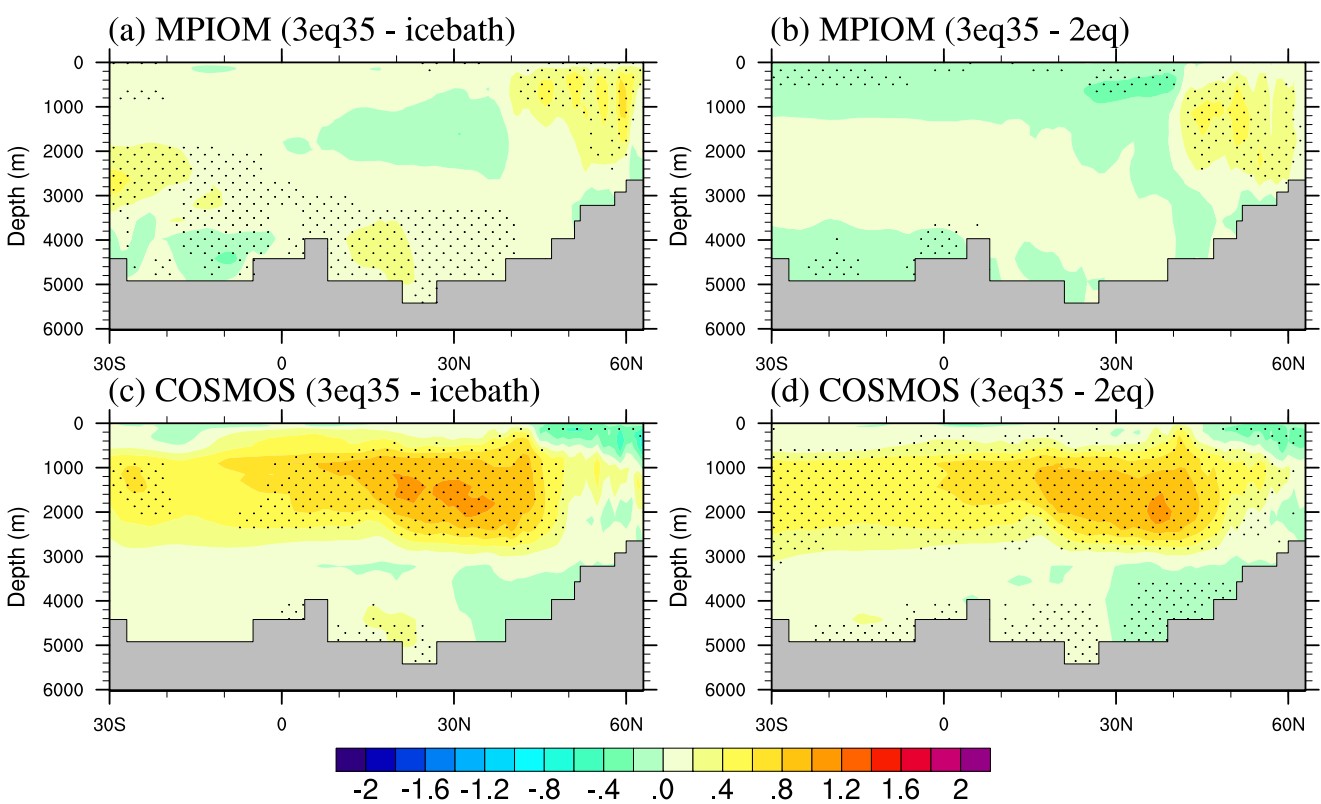

**Figure 8.** Anomalies in AMOC (a) MPIOM-3eq35 – MPIOM-icebath, (b) MPI-3eq35 – MPI-2eq, (c) COSMOS-3eq35 – COSMOS-icebath, and (d) COSMOS-3eq35 – COSMOS-2eq. Units: Sv.

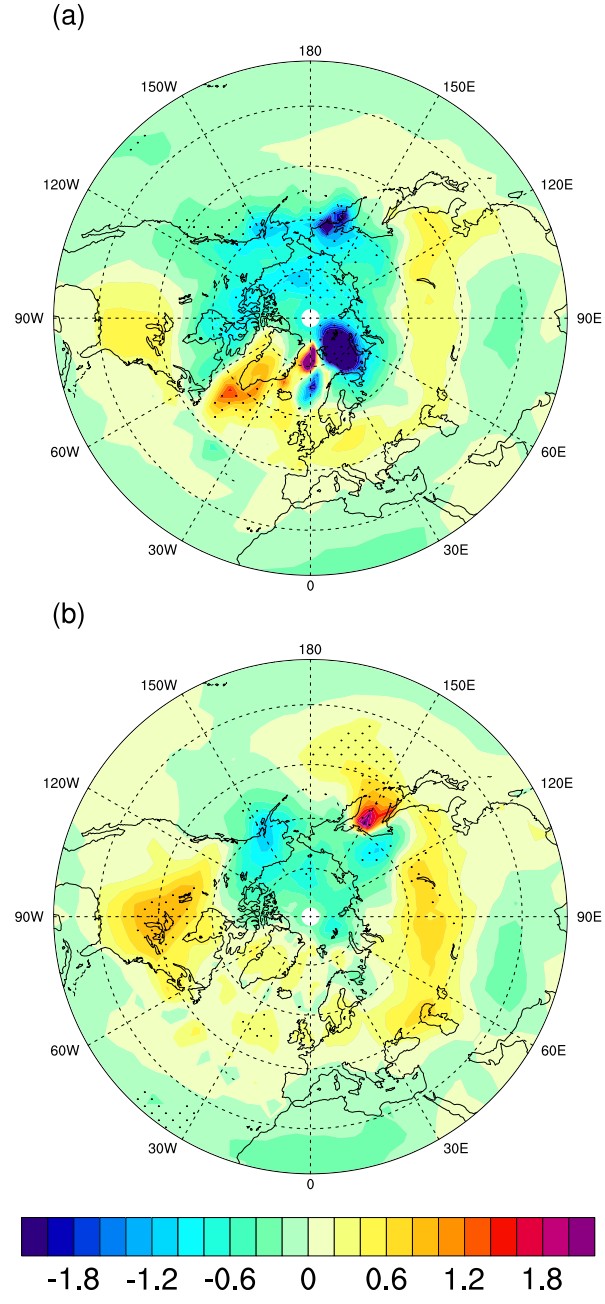

**Figure 9.** Anomalies in surface air temperature (a) COSMOS-3eq35 – COSMOS-icebath, and (b) COSMOS-3eq35 – COSMOS-2eq. Units: K.

**Table 2.** AMOC index

| Experiment | AMOC index | NP index |
|---|---|---|
| MPIOM-icebath | 20.1 | - |
| MPIOM-2eq | 20.2 | - |
| MPIOM-3eq35 | 20.2 | - |
| COSMOS-icebath | 16.6 | 1017.5 |
| COSMOS-2eq | 16.8 | 1017.4 |
| COSMOS-3eq35 | 17.6 | 1017.9 |

The response in surface air temperature, as shown in Fig. 9, indicates a general warming over the Arctic Ocean and its adjacent continents in the COSMOS-icebath and COSMOS-2eq compared to COSMOS-3eq35; a cooling can be found for the Greenland Sea, Nordic Sea, North Atlantic Ocean, southeastern North America, and mid-latitude Eurasia. There are various reasons responsible for these changes: 1) Reduced Arctic sea ice mass in the ice-bath and 2-equation approaches lead to a decrease in the surface albedo, resulting in more heat flux absorbed by the surface. 2) The decline of AMOC in experiments COSMOS-icebath and COSMOS-2eq weakens the northward heat transport from lower latitudes to North Atlantic regions. 3) The atmospheric circulation also plays a role, which is discussed in the following.

Fig. 10 depicts the responses in boreal winter sea level pressure (SLP). Compared to the most realistic parameterization, the simplified approaches illustrate a more negative North Atlantic Oscillation (NAO) mode, with positive SLP anomalies over the Greenland and Nordic seas and negative anomalies over the North Atlantic subtropical zone. SLP anomalies in another time window show similar pattern (Fig. S1), indicating the robustness of the NAO- signal in the simplified approaches, even though the significance level does not exceed 95%. Composite analysis shows that a positive NAO mode leads to a warming over much of Europe and far downstream as the winter-time enhanced westerly flow across the North Atlantic moves relatively warm and moist maritime air to that region (Fig. S2a). Another notable feature is the cooling and warming over North Africa and North America respectively, which is associated with the stronger clockwise flow around the subtropical Atlantic high-pressure center. These described patterns are consistent with the modeled surface air temperature response over Northern Hemisphere continents (Fig. 9).

Another intriguing pattern in the atmosphere is an anomalous negative SLP over the North Pacific Ocean in the simplified parameterizations compared to the most realistic approach. Here we calculate the North Pacific (NP) index as the area-weighted SLP over the region of 30-65°N, 160°E-140°W during boreal winter (Trenberth and Hurrell, 1994). The NP index in COSMOS-3eq35 is shown to be 0.4-0.5 hPa higher than its counterparts (Table 2). A high NP index leads to a warming over southern North America and northern Eurasia, as well as a cooling over northern North America (Fig. S2b), resembling the pattern of the surface air temperature anomalies (Fig. 9). Therefore, the response of the surface air temperature in the simplified parameterizations can be attributed to the combined effect of the weakened AMOC and NAO, and the enhanced Aleutian Low.

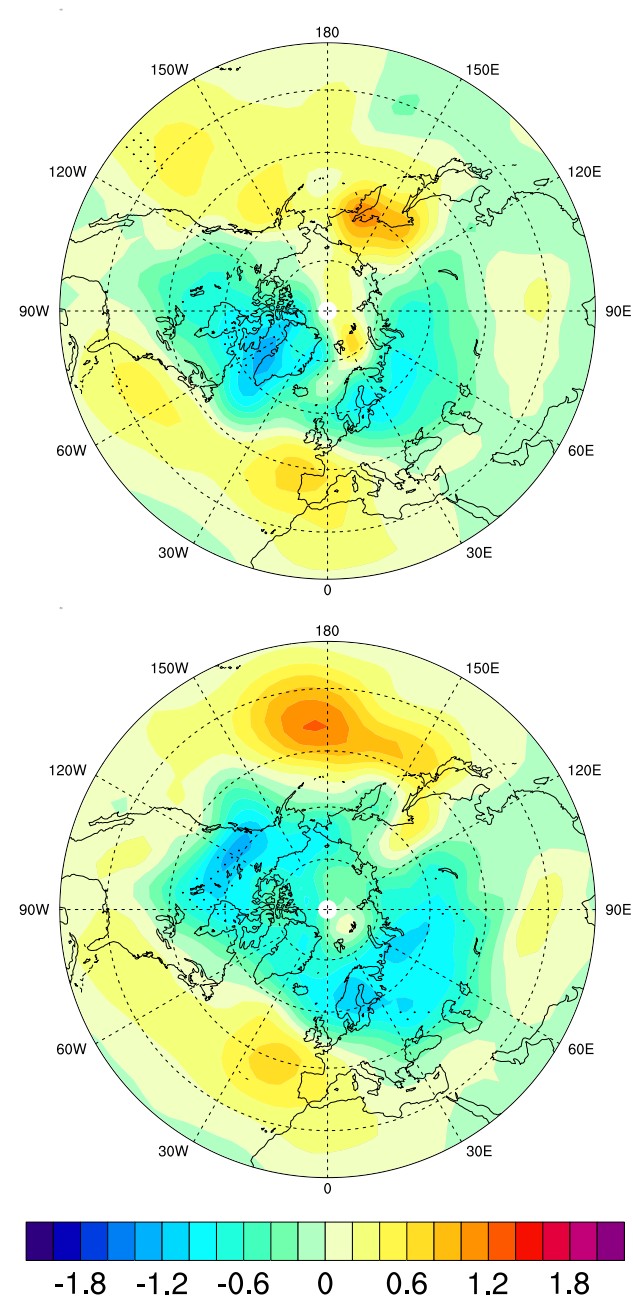

**Figure 10.** Anomalies in boreal winter sea level pressure (a) COSMOS-3eq35 – COSMOS-icebath, and (b) COSMOS-3eq35 – COSMOS-2eq. Units: hPa.

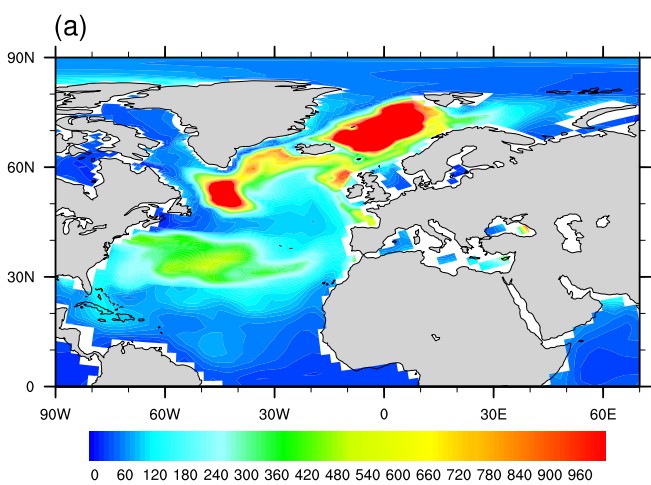

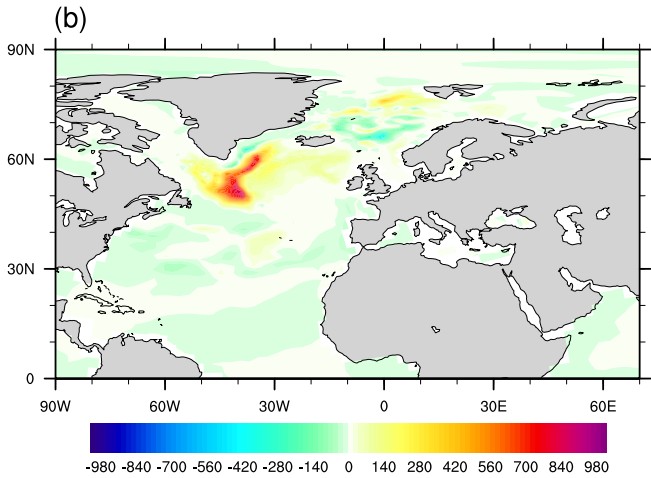

**Figure 11.** (a) Distribution of March mixed layer depth in COSMOS-3eq35. (b) Composite map of mixed layer depth and NAO index for COSMOS-3eq35. It is calculated by averaging March mixed layer depth anomalies (departure from the mean state) during years when the NAO index exceeds one standard deviation. Units are m.

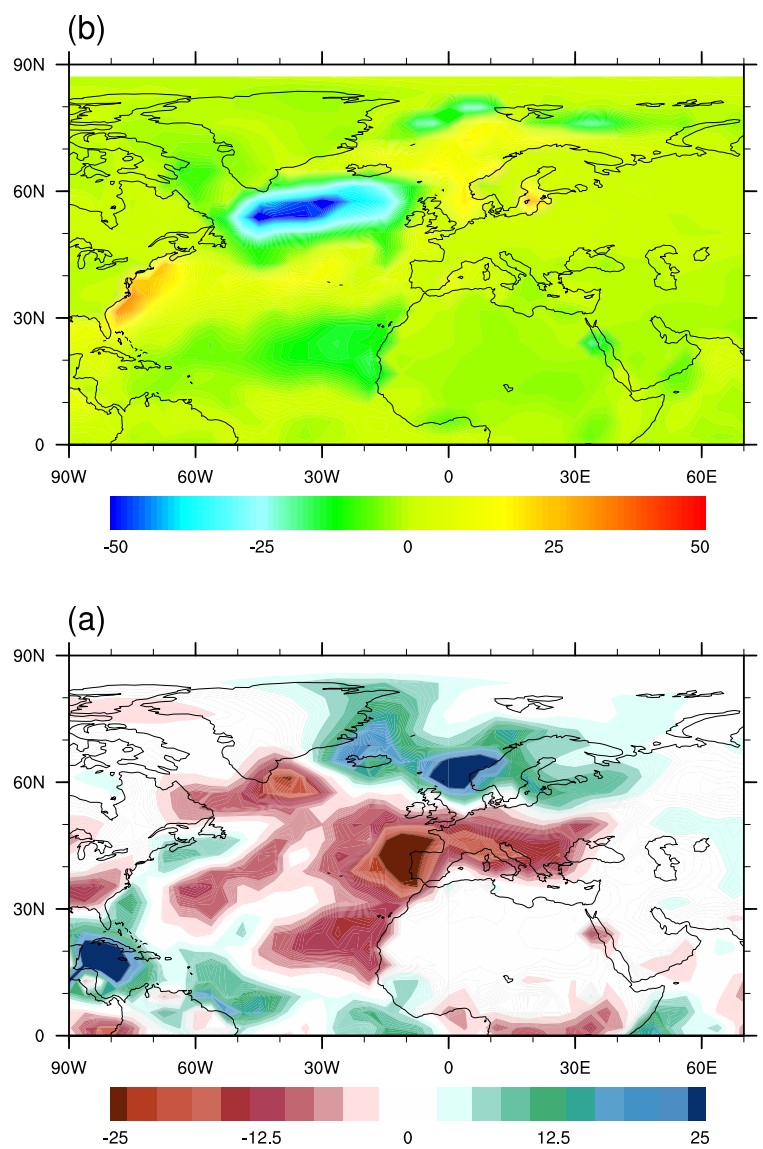

**Figure 12.** Composite map of (a) surface heat flux and (b) net precipitation and NAO index for COSMOS-3eq35. It is calculated by averaging winter anomalies of (a) surface heat flux and (b) net precipitation (departure from the mean state) during years when the NAO index exceeds one standard deviation. Units are W/m$^2$ and m.

## 5.6 Air-sea interaction

In this section the mechanism explaining the weakening AMOC in COSMOS-icebath and COSMOS-2eq as compared to COSMOS-3eq35 is explored. It has long been recognized that the NAO variability has an important influence on the AMOC (Curry et al., 1998; Delworth and Zeng, 2016). Variations in the NAO have been hypothesized to play a role in AMOC variations by modifying air–sea fluxes of heat, water, and momentum. A similar relationship between NAO and the AMOC has been reported also for past climate conditions (Shi and Lohmann, 2016; Shi et al., 2020). Here in Fig. 11 we show the results from a composite analysis between the NAO index and the anomalies in mixed layer depth based on COSMOS-3eq35. It is calculated by averaging March mixed layer depth anomalies (departure from the mean state) during years when the NAO index exceeds one standard deviation.

The convective activities in the Labrador Sea and the Greenland-Iceland-Norwegian (GIN) Seas are shown to have important contributions to the production and transport of North Atlantic deep water (Fig. 11a). For comparison we also show the distribution of mixed layer depth in MPIOM-3eq35 in Fig. S3, which indicates a different location of main deep water convection site in our ice-ocean coupled model: the northeastern North Atlantic. The results from the composite analysis shown in Fig. 11b indicate that the anomalous NAO pattern can lead to significant changes in the ocean circulation. We find that the intensity of the Labrador Sea convection is characterized by variations that appear to be synchronized with variabilities in the NAO. Therefore, the weakening of AMOC in our simplified setups compared to the most realistic approach can be attributed to the simulated anomalous negative NAO phase.

The NAO affects the sea water convection mainly via modifying the surface heat fluxes which leads to anomalies in the spatial and vertical density gradient. Fig. 12a shows the composite map between surface heat flux anomalies and the NAO index. During the positive phase of NAO, more heat than usual is removed from the ocean to the atmosphere at the western Atlantic, in particular the Labrador Sea. Such pattern is in good agreement with the NAO-relative heat flux anomalies derived from the European Centre for Medium-Range Weather Forecasts (ECMWF) interim reanalysis (Delworth and Zeng, 2016). The enhanced removal of heat favors an increase in the surface density and thereby strengthens deep water formation. On the other hand, the NAO also affects the net precipitation over North Atlantic Ocean. As illustrated in Fig. 12b, relatively drier condition could occur over Labrador Sea and Irminger Sea during positive NAO years.

## 6 Discussion and conclusion

In the present study, we perform 1-D simulations with an idealized columnar model (SIM), as well as global simulations with a stand-alone sea ice model (CICE), an ice-ocean coupled model (MPIOM), and a fully coupled climate model COSMOS, to analyze the sensitivity of modeled climate to ice-ocean interface heat flux parameterizations. This is achieved by implementing into the models: 1) a simple ice-bath assumption with the ocean temperature fixed at the freezing temperature, 2) a more realistic bulk 2-equation with freezing temperature kept at the ice–ocean interface and the ocean being allowed to be warmer than freezing point (McPhee, 1992) and 3) a most advanced double diffusional transport (3-equation) approach with the temperature at the ice–ocean interface being calculated based on the melting rate of the ice bottom (Notz et al., 2003).

The conclusions drawn from these models in terms of sea ice properties are quite similar with each other: The thinnest ice is observed in the ice-bath simulations, as no residual heat is allowed to remain in the ocean and the sea water beneath sea ice is constantly at its freezing point. The 2-equation experiments simulate thicker sea ice, because some of the heat is stored in the ocean rather than used for ablating the ice. The simulated sea ice by the 3-equation approach has the largest thickness, as the temperature at the ice-ocean interface can exceed the freezing point of the far-field ocean, causing the heat flux from the ocean to be reduced or even reversed. The marginal ice areas are found to be highly sensitive to the choice of ice-ocean heat flux parameterizations. In particular, the sea water temperature in the marginal ice zones is largely determined by the onset/retreat of the sea ice.

As a result of the brine release during sea ice formation, the Arctic Ocean is most salty in the 3-equation experiment and least salty in the ice-bath experiment; the same is found in the deep water masses due to their coupling with the surface source water. The thermohaline instability obtained from such salinity profile is responsible for a strengthening of the Atlantic meridional overturning circulation (AMOC) in the coupled simulation with the 3-equation approach. Note that our results are in good agreement with a previous study using CICE (Tsamados et al., 2015) that found in August stronger basal -melting of Arctic sea ice, decreased Arctic Ocean salinity, cooling of sea water in the central Arctic and warming of sea water at the ice edge in the 2-equation experiments compared to the 3-equation approach. However, in their study the effects are more pronounced, possibly because we used different model versions of CICE and different parameters for the ice-ocean heat flux formulations, one example is the value for R, which is 50 in Tsamados et al. (2015) and 35 in our case. In addition, different atmospheric forcings were used in the two studies.

In contrast to their and other previous studies, in our study we do not only use a stand-alone sea-ice model but also analyse a coupled ice-ocean model and an earth-system model. These allow us to examine the effect of various oceanic heat-flux formulations on the deep ocean and the atmospheric circulation as well as their impact on sea-ice properties. In our study, COSMOS reveals intensification in both the AMOC and NAO when the most advanced ice-ocean heat flux parameterization is applied. Ocean observations and model simulations show that the changes in the thermohaline circulation during the last century have been driven by low-frequency variations in the NAO via changes in Labrador Sea convection (Latif et al., 2006). More recently, a delayed oscillator model as well as a climate model suggest that the NAO forces the AMOC on a 60-year cycle (Sun et al., 2015). The strengthening of the AMOC, obtained in our COSMOS-3eq experiment, is likely due to the combined effect of increased thermohaline instability and the amomalous NAO+ mode. In contrast, no obvious response of the AMOC can be found in the MPIOM experiments (Table 2). As indicated in the present paper and many other studies (Curry et al., 1998; Latif et al., 2006; Sun et al., 2015), thee AMOC is closely related to the atmospheric processes over North Atlantic Ocean. One of the key elements controlling the atmospheric circulation over the North Atlantic is the NAO. As the atmospheric forcings are prescribed in MPIOM, there is no difference in the atmospheric state among the MPIOM experiments. Therefore, the prescribed atmospheric forcing largely limits the air-sea interaction feedback.

Our study indicates a less pronounced sea-ice response to ice-ocean interface heat flux parameterizations in the fully coupled climate model COSMOS than in the ice-ocean model MPIOM (compare Fig. 3e-h with Fig. 3i-l). This is because the change of the AMOC has a dampening effect on the simulated sea ice anomalies. The strengthening of the AMOC in COSMOS-3eq

can lead to a warming over the Northern Hemisphere, especially over the North Atlantic and the Arctic. This hypothesized link between the AMOC and Northern Hemisphere mean surface climate has been documented in an abundance of studies (e.g., Schlesinger and Ramankutty, 1994; Rühlemann et al., 2004; Dima and Lohmann, 2007; Parker et al., 2007). The AMOC-induced warming helps to reduce the sea ice mass over the Arctic and North Atlantic subpolar regions. Indeed, the sea ice across the Greenland Sea and Baffin Bay are found to be thinnest in COSMOS-3eq.

It should be noted that CICE is in many aspects different from the sea ice component in MPIOM: 1) CICE uses the multi-layer approach with a sub-grid scale ice-thickness distribution (Bitz and Lipscomb, 1999), while MPIOM uses zero-layer thermodynamics following Semtner Jr (1976). 2) A submodel of ice dynamics based on an elastic-viscous-plastic rheology (Hunke and Dukowicz, 1997; Hunke, 2001) is used in CICE, while in MPIOM viscous-plastic dynamics following Hibler III (1979) are used. 3) Different spatial resolutions are used in CICE ($1°$) and MPIOM ($3°$). 4) CICE is forced by monthly-mean

climatological data from National Center for Atmospheric Research (NCAR), while the MPIOM experiments are forced by daily fields from the climatological OMIP data set (Röske, 2006). Therefore, the different model behavior between CICE and MPIOM can to a certain extent be explained by the different model configurations.

A detailed comparison of the simulations carried out here with observational data is beyond the scope of our study. However, we note that the sea ice thickness simulated by COSMOS have been evaluated by Notz et al. (2013), who found an

460 overestimation of Arctic sea-ice thickness in ECHAM5/MPIOM (i.e., COSMOS) with the ice-bath formulation compared to the reanalysis from the Pan-Arctic Ice-Ocean Modeling and Assimilation System (PIOMAS) in both winter and summer. Improving the formulation of the ice–ocean heat flux by applying the 3-equation approach causes thicker ice and hence further increases this particular model bias. This indicates that the simplified heat flux parameterisation partly compensates for other errors in the coupled model setup.

In the present paper, we exclude the responses of the Southern Ocean, as these are much less pronounced than those of the Arctic Ocean. Another reason lies on the over-estimation of the Southern Ocean sea ice extent by the stand-alone sea ice model CICE due to a lack of represented warm deep water.

Our study provides a better understanding of the impact of a realistic representation of ice-ocean heat flux processes in large scale-climate models, including their effect on sea ice, ocean circulation, and the atmosphere. We find that substantial, large

scale climate metrics can emerge from the different parameterization, highlighting the importance of a careful evaluation of their impact in climate-model simulations.

*Code and data availability.* The source code, data as well as scripts for plotting the figures in this manuscript can be downloaded from https://doi.org/10.5281/zenodo.5036700.

*Author contributions.* D. Notz, J. Liu and X. Shi developed the original idea for this study. X. Shi and H. Yang contribute to the code modification and model simulation under the supervision of J. Liu (for CICE), D. Notz (for SIM and MPIOM), and G. Lohmann (for COSMOS). All authors contributed to the data analysis, discussion and paper writing.

*Competing interests.* The authors have declared that no competing interests exist.

*Acknowledgements.* This research is supported by the National Key R&D Program of China (2018YFA0605901) and NSFC (41676185); the Institute of Atmospheric Physics, Chinese Academy of Sciences (IAP/CAS); Max Planck Institute for Meteorology (MPI-M); Alfred Wegener Institute, Helmholtz center for Polar and Marine Research; PACMEDY of the Belmont Forum; the second phase of PALMOD project, and the open fund of State Key Laboratory of Loess and Quaternary Geology, Institute of Earth Environment, CAS (SKLLQG1920). D. Notz is funded by the Deutsche Forschungsgemeinschaft under Germany's Excellence Strategy EXC 2037 'CLICCS - Climate, Climatic Change, and Society' Project Number: 390683824, contribution to the Center for Earth System Research and Sustainability (CEN) of Hamburg University. We would to express our appreciation for the constructive comments from Prof. Bruno Tremblay and another anonymous reviewer. In addition, we would like to thank Karl-Hermann Wieners, Helmuth Haak, Xiucheng Wang and Mirong Song for their technical help with the MPIOM and CICE models, and thank Nils Fischer for very interesting discussions.

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
