# Peer review of "Sensitivity of Northern Hemisphere climate to ice-ocean interface heat flux parameterizations"

_Geoscientific Model Development, 2020_

## Referee Comment (RC1) · Bruno Tremblay (Referee) · 28 Dec 2020

**A review of Sensitivity of Northern Hemisphere climate to ice-ocean interface heat flux parameterizations**
**X. Shi, D. Notz, J. Liu, H. Yang and G. Lohmann**

**1   Summary**

This paper presents a study of the impact of three increasingly realistic parameterizations of the ice-ocean turbulent heat flux (ice-bath, 2-equation,3-equation) on the seasonality of Arctic sea ice (thickness and concentration) and the climate system in general, using four different models with increasing level of complexity (SIM, CICE, MPIOM, COSMOS). In the ice-bath model, the top ocean grid cell is simply fixed at freezing temperature. The 2-equation and 3-equation models have a linear dependency on the friction velocity and the temperature difference between ocean mixed layer and its interface with the ice. Their difference lies in the definition of the interface temperature: the freezing temperature of the top grid cell of the ocean or the freezing temperature of the water at the interface which depends on the ice bottom melting rate. Results show that the simulation of the seasonal cycle in sea ice thickness and concentration improves with the complexity of the model. Results also show that the spatial distribution of sea surface temperature is insensitive to the treatment of the ice-ocean turbulent heat flux in winter and summer, except at the ice edge in summer for the 3-equation model. In the Arctic Ocean, the 3-equation model leads to cooler deep waters and saltier waters over the whole column. This results in a stronger NAO and AMOC. Based on these results, the authors argue for the importance of a realistic parameterization of ice-ocean heat exchange.

The paper presents an insightful and detailed analysis of the impact of different treatment of the ice-ocean turbulent fluxes on Arctic climate simulation. The introduction, however, lacks details about the relevance of the study and how it fits in the context of previous studies, as well as with the presentation of the three parameterizations. The paper is generally well organized and clear but it should be proof-read for English grammar – particularly after the Introduction section. We recommend that the paper be accepted for publication after the comments below have been addressed.

**2   Major comments**

a Abstract: ”...a similar turbulent heat-flux parameterization as (2) but with the temperature at the ice-ocean interface depending on ice-ablation rate”. The general reader will not appreciate this sentence early in the paper, i.e. before having read the rest of the paper. An explicit reference to the three-equation model should be included here.

b Abstract: The abstract should report on all key results. As it stands, only results from Model (3) are reported and one wonders why option (1) and (2) were considered in the first place if they don't deserve a line in the abstract.

c l19, Introduction: The same Schmidt et al., 2004 reference is used for the simplest parameterization (fixed Tocn) and also for the most complex parameterization (3-eqs model). This is confusing. The contribution of Schmidt et al. is the system of 3-eqs to solve for the interface temperature, not the fixed Tocn at freezing point. Another (earlier) reference should be used for the fixed Tocn parameterization.

d The authors should describe how their work fits in with the existing literature in both the Introduction and Discussion. Tsamados et al., 2015 examines similar questions. How do the results of this study compare with those of Tsamados et al? What is learned from this study that was not known before?

e The background provided in the introduction is well written. However, the overall motivation for the study and its relevance to the field is not clearly stated. A discussion of how different climate models currently simulate ice-ocean heat fluxes would go a long way in addressing this point. This is stated at the end of the paper. It should also be stated earlier in the paper.

f The presentation of results from the ice-bath parameterization should be motivated given that the authors states that it is "incompatible with observations" (l.24)?

g l.45. The introduction should include a discussion of the different ways of calculating the ice-ocean heat flux for all three methods. This is only clarified in section 2. This should also come earlier in the paper, in section 1.

h l.170: Ideally, the experiments would be done with the same GCM, subsequently removing components to reduce the level of complexity. As it stands now, the CICE model has an ITD, when the GCM (COSMOS) does not. The same comment applies for the forcing that changes between models. This would facilitates the comparison of simulations with different model components. As it stands now, we are left wondering how much of the difference in behavior between models is due to the different model components and forcing rather than the ice-ocean turbulent flux parameterization. GCMs typically has this functionality. If COSMOS does not have this capability, it should be acknowledged, and this caveat should be mentioned.

i Section 2, l92-110: The discussion of the methods, the caveats related to each method, and how each method relate to previous studies should be streamlined. References should appear in parentheses for clarity and text that does not pertain to the Heat Flux Parameterisations should appear in other sections (e.g. Discussion).

j l.206. The temperature at the interface for SIM-icebath is described here; yet it is not used in ice-bath. Are $T_f$ and $T_{interface}$ from Equ. (2) and (3) for the ice-bath and 2-equation parameterizations (respectively) always the same? Does $T_f$ for the ice-bath parameterization also depend on salinity? If so, that would also make it a "2 equations" problem. The differences between each "temperatures" in all three parameterizationa should be described more clearly. A table with the formulas and temperatures used in each method would help clarify this issue.

k l.99 Why is the sensitivity of the model to the parameter R tested giventhat "R=35 best describes reality"?

l Section 4.1: This is where the authors examine the sensitivity of the choices described in Section 2 using an idealized model. A more detailed discussion of these results (along with additional figures) should be included in this section.

m Section 5: One question that keeps coming up in the readers mind while reading the results section is: did the model become more realistic as a result including a more realistic ice-ocean heat flux? Sometimes, especially in complex models, fixing one thing can often expose other problems, perhaps related to model tuning. With respect to this, the authors might consider describing how the results using the most realistic ice-ocean flux parameterization compare with observations. Sea ice thickness is tricky to measure, but Labe et al., 2018 could be a good start (They compare CESM, which uses CICE, to PIOMAS data). For the ocean surface, a qualitative comparison could be made to Peralta-Ferriz and Woodgate 2015, which is a nice study looking pan-Arctic ocean observations over the past 30 years.

n l218 and lines below. l.218 should read "difference in annual mean ice thickness increases with friction velocity". The same form should be used for l220 and other instances that appear below: e.g. "... the larger the deeper our mixed layer is". The same comment for "mixed-layer and atmospheric temperature", and again for "ice concentration and ice thickness".

Line 225: Do not word "explicable" should not be used in this context. Use instead: "This is explained by the fact that.."

**3   Minor comments**

a l.9, abstract. 'The most realistic representation". This is vague. It should reference parameterization (1), (2) or (3) above.

b l.49-51. grammatical errors.

c l.53-57. This sentence should be broken into more than one sentence for clarity.

d l.71. $T_{mix}$ is defined as the ocean temperature. The vertical level (e.g. first layer) should be stated since the temperature is not constant in the mixed layer of CICE.

e l.80. The freezing-point equation for seawater should be written explicitly.

f l.100. typo: salt and heat are transported almost equally efficiently.

g Section 3: There are several instances where numbers appear in the text or within equations without references. The references should be added.

h l.121. $T_s$ is the ICE surface temperature. This should be defined. $T_{bot}$ should also be defined.

i l.116. The acronym MPIOM should be defined when it first appear.

j l.127: A reference for $k_i$ should be given. Is this measurement really precise up to two decimal places?

k l.130 The Notz et al. and Maykut and Untersteiner references should appear above on line 129, i.e. before Fsw nd Fother is introduced.

l (5), l.130,l.131, l.135. The number in the equations should be replaced by symbols; the "$\times$" should be removed and the numerical values should appear in the text, for the sake of clarity: (eg. $5.67 \times 10^{-8}$ would be $\sigma$)

m l.126. The bulk freezing temperature of the ice should be defined.

n l.130. Numbers should be added for equations that appear after (5).

o l.147: "the so-called CCSM3 set-up". This is not a commonly know term. The set-up should simply be defined. There are a lot of acronyms in the paper that are used and not defined. Acronyms should be defined when they first appear.

p l.153 Is the mixed-layer salinity held constant in all the models? This would have an impact on the freezing temperature, and therefore on the heat flux parameterizations. This should be discussed (though not necessarily in the methods section).

q l.154 What climatological dataset is being referred to exactly? The reference to the dataset should be included here. The time period over which the climatology was calculated should also be stated.

r l.166: The ice distribution parameter must be described. The paper should be stand-alone.

s l.188, l.193: "We start with" is used twice.

t l.191: "which is set to either 35 or 70". This is redundant as the simulations are listed above.

u l.192: "no more changes from one year to the next." What were the initial changes?

v l.202: Why is the mixed-layer warming more slowly in SIM-2eq and 3eq? Is it that the ice cover is lost earlier when the sun is higher over the horizon? This should be stated.

w l.208: The word "faster" should be used instead of "stronger" when describing "ice melt".

x Section 4.2: This subsection consists of a single paragraph that is approximately one page long. It should be broken into several paragraphs for clarity.

y l.240. The student's t-test should be more clearly described. Is it testing that the results from two different parameterizations are significantly different?

z Figures: Units and labels should be included on all figures.

Fig.1. Units are missing.

Fig.3-4. A larger font size for longitudes and the colorbars should be used.

Fig.6-7. Units are missing for depth on the y axis.

Fig.6-7. The seabed should appear in grey or a different color to differentiate it from zero in the water.

Table 1: The parameter R should be defined/included in the Table caption. Currently, it is only defined later on l.192

References:

- Tsamados, M., Feltham, D., Petty, A., Schroder, D. & Flocco, D. Processes controlling surface, bottom and lateral melt of Arctic sea ice in a state of the art sea ice model. Philos. Trans. R. Soc. A 17, 10302 (2015).

- Labe, Z., Magnusdottir, G. & Stern, H. Variability of Arctic sea ice thickness using PIOMAS and the CESM large ensemble. J. Clim. 31, 3233–3247 (2018).

- Peralta-Ferriz, C. & Woodgate, R. A. Seasonal and interannual variability of pan-Arctic surface mixed layer properties from 1979 to 2012 from hydrographic data, and the dominance of stratification for multiyear mixed layer depth shoaling. Prog. Oceanogr. 134, 19–53 (2015).

Bruno Tremblay, McGill University
Jointly with MSc student and Postdoctoral Fellow

---

## Referee Comment (RC2) · Anonymous Referee #2 · 5 Mar 2021

Review of Sensitivity of Northern Hemisphere climate to ice-ocean interface heat flux parameterizations by Shi et al

This paper presents the impact of various ice-ocean heat flux parameterizaton on several aspects of the climate of the Northern Hemisphere within four models of increasing complexity. The paper is clearly laid out. I find the analysis ambitious and interesting but would need some clarifications to be satisfied of its robustness and significance. I am also curious as to the chosen focus on this process (among so many others). I suggest major corrections (see below) before the paper can be accepted for publication.

General comments:

1) The fixed depth mixed layer model in the 1D and stand-alone models is a clear simplification that could affect the key results. For example a thinner mixed layer warms up more under fragmented ice in summer and I expect this to really influence your conclusions. Please discuss and provide additional information. 2) I would like clarification on how the ice-bath model can be implemented in the ice-ocean coupled models. 3) Elaborate on the mechanisms that explain the weakening of the THC. 4) How does the mixed layer look like in the ice-ocean coupled models (i.e. depth, ...) 5) Scales on figures are chosen to essentially show the sign but not so much the magnitude of the differences (i.e, one can tell where ice is thicker or thinner but not by how much). Is that to hide the large differences that cannot be easily explained between model setups? 6) If the mixed layer temperature is so critical in controlling the temperature of the deep waters then it is all the more important to give a convincing description of its evolution and realism 7) A comparison with Tsamados et al (2015) would be useful especially as the author of this study found a small impact of the 3eqBC. Discuss 8) Is the most advanced thicker because of the reduced (even reversed) summer fluxes from the ocean to the ice? Again why is this results not so marked in Tsamados et al (2015) 9) I am really uncertain as to the significance of the impacts found on the ocean and atmosphere. How does this compare to internal variability within the models? I have heard in the past Notz state that sea ice physics does not play a significant role in the climate response (I might be misquoting and apologies if I do) but how do these finding square with this view? 10) Are the results presented reproducible. What if you analysed another 10 years or 100 years period? 11) I wonder why you do not use the same setup to analyse several other sea ice processes (albedo, melt pond, form drag, as per the recipe of Tsamados et al 2015 etc. ...). Is it too good to be true? 12) Not clear how the prescribed atmospheric forcing subdues the impact on AMOC. Please elaborate.

Specific comments:

P1 L18 expand on motivation and justification P1 L20 clarify this paragraph. Which freezing temperature... P1 L27 Here is a good place to expand on the analogy and

differences between momentum and heat transfer and write the equations and if need be criticise what is missing in either of them. At the moment it is too vague. For example what do you 'differs from the exchange of momentum" in what way? P2 L42 a local phase P2 L50 , Tsamados et al (2015) P4 L85 together with the freezing equation (1) P4 L 91 2003) and as implemented in CICE by Tsamados et al (2015) P5 L137 fixed mixed layer depth? P6 L141 version? Expand + maybe P6 L150 again default mixed layer of fixed depth. Not realistic, this affects your Tmix and hence your results P6 L153 same issue with salinity should change with mixed layer depth P6 L160 expand P6 L165 The repartition? Rephrase slightly P6 L169 wouldn't it better to have it at 1 deg and run for 100 years? Or is that needed for equilibration? P8 L187 and ice concentration P8 L210 more slowly P10 L221 a smaller ... for a deeper ... P10 L226 are larger ... P10 L238 I am surprised that you needed a 90 years spin-up for a stand-alone sea ice model P13 L257 what do you mean by far-field? Mixed layer? Also typo...at the interface P13 L268 I am not clear on how you can obtain an ice bath situation in the ice-ocean coupled models P13 L269 how significant is this cooling in COSMOS? How does it compare to model internal variability for example? Fig5 why don't you show CICE? P13 L274 doesn't less ice mean more growth in winter (negative feedback) and hence more brine release? P13 L280 the small differences between COSMOS-2eq and COSMOS-3eq35 indicates that once mixed layer allowed to evolve impact of this parameterisation is small? P13 L278 I don't get this explanation P17 L288 where are these regions of deep water formation? P17 L289 I am not sure I follow why thinner means fresher mixed layer. The system rapidly equilibrates to a thinner state and then no reason to have fresher ML P17 L293 could it be because these results are coincidental Fig S2 caption -> departure from

---

## Author Response (AR1)

**Response letter to Reviewer 1**

Xiaoxu Shi[1], Dirk Notz[2,3], Jiping Liu[4], Hu Yang[1], and Gerrit Lohmann[1]

[1]Alfred Wegener Institute, Helmholtz center for Polar and Marine Research, Bremerhaven, Germany
[2]Institute for Oceanography, Center for Earth System Research and Sustainability (CEN), Hamburg University, Hamburg, Germany
[3]Max Planck Institute for Meteorology, Hamburg, Germany
[4]Department of Atmospheric and Environmental Sciences, University at Albany, New York, USA

**1  Summary**

1 This paper presents a study of the impact of three increasingly realistic parameterizations of the ice-ocean turbulent heat flux (ice-bath, 2-equation, 3-equation) on the seasonality of Arctic sea ice (thickness and concentration) and the climate system in general, using four different models with increasing level of complexity (SIM, CICE, MPIOM, COSMOS). In the ice-bath model, the top ocean grid cell is simply fixed at freezing temperature. The 2-equation and 3-equation models have a linear dependency on the friction velocity and the temperature difference between ocean mixed layer and its interface with the ice. Their difference lies in the definition of the interface temperature: the freezing temperature of the top grid cell of the ocean or the freezing temperature of the water at the interface which depends on the ice bottom melting rate. Results show that the simulation of the seasonal cycle in sea ice thickness and concentration improves with the complexity of the model. Results also show that the spatial distribution of sea surface temperature is insensitive to the treatment of the ice-ocean turbulent heat flux in winter and summer, except at the ice edge in summer for the 3-equation model. In the Arctic Ocean, the 3-equation model leads to cooler deep waters and saltier waters over the whole column. This results in a stronger NAO and AMOC. Based on these results, the authors argue for the importance of a realistic parameterization of ice-ocean heat exchange.

The paper presents an insightful and detailed analysis of the impact of different treatment of the ice-ocean turbulent fluxes on Arctic climate simulation. The introduction, however, lacks details about the relevance of the study and how it fits in the context of previous studies, as well as with the presentation of the three parameterizations. The paper is generally well organized and clear but it should be proof-read for English grammar – particularly after the Introduction section. We recommend that the paper be accepted for publication after the comments below have been addressed.

**Dear Reviewer,**

**Thank you very much for your constructive comments. In the following, we present our point-to-point responses. Our answers to your comments are written in bold.**

**Thanks again for your time and efforts.**

**Best, Xiaoxu**

**2 Major comments**

25 a Abstract: "...a similar turbulent heat-flux parameterization as (2) but with the temperature at the ice-ocean interface depending on ice-ablation rate". The general reader will not appreciate this sentence early in the paper, i.e. before having read the rest of the paper. An explicit reference to the three-equation model should be included here.

**We now take direct reference to the three-equation model as follows:**

30 *These three parameterizations are (1) an ice-bath assumption with the ocean temperature fixed at the freezing temperature; (2) a 2-equation turbulent heat-flux parameterization with ice-ocean heat exchange depending linearly on the temperature difference between the underlying ocean and the ice-ocean interface whose temperature is kept at the freezing point of the sea water; and (3) a 3-equation turbulent heat-flux approach in which the ice-ocean heat flux depends on the temperature difference between the underlying ocean and the ice-ocean interface whose temperature is calculated based on the local salinity set by the ice-ablation rate.*

35 b Abstract: The abstract should report on all key results. As it stands, only results from Model (3) are reported and one wonders why option (1) and (2) were considered in the first place if they don't deserve a line in the abstract.

**We referred to the most realistic parameterization (3) as our reference/control simulation, therefore we compared the simulation results of (1) and (2) to (3). Based on the reviewer's comment, in our abstract we now include a discussion of the differences between (1) and (2), so the reader can obtain a more detailed overview of our key results.**

40 **The following texts can also be found in the revised manuscript at line 7-16.**

*Based on model simulations with the standalone sea-ice model CICE, the ice-ocean model MPIOM and the climate model COSMOS, we find that compared to the most complex parameterization (3), the approaches (1) and (2) result in thinner Arctic sea ice, cooler water beneath high-concentration ice and warmer water towards the ice edge, and a lower salinity in the Arctic Ocean mixed layer. In particular, parameterisation (1) results in the smallest sea ice thickness among the 3*
45 *parameterizations, as in this parameterisation all potential heat in the underlying ocean is used for the melting of the sea ice above. For the same reason, the upper ocean layer of the central Arctic is cooler when using parameterisation (1) compared to (2) and (3). Finally, in the fully coupled climate model COSMOS, parameterisations (1) and (2) result in a fairly similar oceanic or atmospheric circulation. In contrast, the most realistic parameterization (3) leads to an enhanced Atlantic meridional overturning circulation (AMOC), a more positive North Atlantic Oscillation (NAO) mode and a weakened*
50 *Aleutian Low.*

c l19, Introduction: The same Schmidt et al., 2004 reference is used for the simplest parameterization (fixed Tocn) and also for the most complex parameterization (3-eqs model). This is confusing. The contribution of Schmidt et al. is the system of 3-eqs to solve for the interface temperature, not the fixed Tocn at freezing point. Another (earlier) reference should be used for the fixed Tocn parameterization.

55 **We now generically refer to a discussion of all three approaches in Holland, 1999; Jenkins, 2001; Notz et al., 2003 and Schmidt et al., 2004 as follows:**

*"A number of approaches exist for the calculation of the oceanic heat flux $F_{oce}$ (c.f. Holland and Jenkins, 1999; Jenkins et al., 2001; Notz et al., 2003; Schmidt et al., 2004). For the simplest parameterization, the ice–ocean system is simply treated as an ice bath: The temperature of the uppermost ocean grid cells is fixed at its freezing temperature, and any excess energy*
60   *that enters these grid cells via advection, convection, or heat exchange with the atmosphere is instantaneously applied to the ice through lateral and bottom melting. "*

d The authors should describe how their work fits in with the existing literature in both the Introduction and Discussion. Tsamados et al., 2015 examines similar questions. How do the results of this study compare with those of Tsamados et al? What is learned from this study that was not known before?

65   **In introduction we add (line 92-105):**

*Exploring the behavior of different parameterizations describing ice-ocean heat flux has been an important topic in model studies. Significant differences can generally exist between melt rates calculated with the 3-equation approach and less realistic approaches (c.f. Notz et al., 2003; Tsamados et al., 2015), as only the 3-equation approach allows for heat fluxes that are directed from the interface into the water and therefore allows for a realistic limitation of melt rates through the*
70   *formation of a fresh water layer underneath the ice. In a recent study the sensitivity of sea ice simulation to the approaches introduced in McPhee (1992) and Notz et al. (2003) have been examined using a stand-alone sea ice model CICE (Tsamados et al., 2015). CICE uses a simple thermodynamic slab ocean with fixed mixed layer depth and sea water salinity. Thus, the realistic effect of oceanic processes can not be represented. For example the sea ice over the Southern Ocean is severely overestimated by CICE due to a lack of warming effect from the Antarctic deep water. Therefore it is necessary to also*
75   *investigate the ice-ocean heat flux formulations in a more complex system, including an interactive ocean or even the atmosphere. Based on this motivation, in the present study, we examine how different physical realism that is represented by the three discussed parameterizations impact the resulting ice cover, large-scale oceanic circulation, and atmosphere properties in different numerical models ranging from an idealized columnar model, a stand-alone sea ice model, an ice-ocean coupled model, to a most complex climate system model.*

80   **In discussion we add (line 400-409):**

*Note that our results are in good agreement with a previous study using CICE (Tsamados et al., 2015) that found in August stronger basal -melting of Arctic sea ice, decreased Arctic Ocean salinity, cooling of sea water in the central Arctic and warming of sea water at the ice edge in the 2-equation experiments compared to the 3-equation approach. However, in their study the effects are more pronounced, possibly because we used different model versions of CICE and different*
85   *parameters for the ice-ocean heat flux formulations, one example is the value for R, which is 50 in Tsamados et al. (2015) and 35 in our case. In addition, different atmospheric forcings were used in our studies.*

*In contrast to their and other previous studies, in our study we do not only use a stand-alone sea-ice model but also analyse a coupled ice-ocean model and an earth-system model. These allow us to examine the effect of various oceanic heat-flux formulations on the deep ocean and the atmospheric circulation as well as their impact on sea-ice properties.*

90 e The background provided in the introduction is well written. However, the overall motivation for the study and its relevance to the field is not clearly stated. A discussion of how different climate models currently simulate ice-ocean heat fluxes would go a long way in addressing this point. This is stated at the end of the paper. It should also be stated earlier in the paper.

**One motivation of our study is to test the 3 heat-flux formulations in sea ice models. Another motivation, as our response to your last comment, arise from the fact that most former studies used stand-alone sea ice models, there is a**

95 **lack of investigation on the oceanic and atmospheric responses, here we refer to our response to comment d.**

**Besides, according to the reviewer's comment, we further add in the introduction the following content (line 106-109):**

*Another motivation of our study is to help improve the formulation describing ice-ocean heat flux in various models. For example in the 4th version of CICE, only ice-bath and 2-equation assumptions could be applied. In MPIOM and COSMOS, the ice-bath approach is used, which can lead to an overestimation of oceanic heat flux into sea ice. In our study*

100 *we implemented the more realistic 3-equation parameterization into all the models mentioned above.*

 f The presentation of results from the ice-bath parameterization should be motivated given that the authors states that it is" incompatible with observations" (l.24)?

**Yes, since this parameterization is incompatible with observations but still used in MPIOM and COSMOS, we would like to improve the heat-flux formulation in the two models. We have included this point in the introduction (the same**

105 **as our response to your last comment) (line 106-109):**

*Another motivation of our study is to help improve the formulation describing ice-ocean heat flux in various models. For example in the 4th version of CICE, only ice-bath and 2-equation assumptions could be applied. In MPIOM and COSMOS, the ice-bath approach is used, which can lead to an overestimation of oceanic heat flux into sea ice. In our study we implemented the more realistic 3-equation parameterization into all the models mentioned above.*

110 g l.45. The introduction should include a discussion of the different ways of calculating the ice-ocean heat flux for all three methods. This is only clarified in section 2. This should also come earlier in the paper, in section 1.

**According to the reviewer's comment, we have merged section2 into section1. Here we refer to the Introduction part in the updated manuscript.**

 h l.170: Ideally, the experiments would be done with the same GCM, subsequently removing components to reduce the

115 level of complexity. As it stands now, the CICE model has an ITD, when the GCM(COSMOS) does not. The same comment applies for the forcing that changes between models. This would facilitates the comparison of simulations with different model components. As it stands now, we are left wondering how much of the difference in behavior between models is due to the different model components and forcing rather than the ice-ocean turbulent flux parameterization. GCMs typically has this functionality. If COSMOS does not have this capability, it should be acknowledged, and this caveat should be mentioned.

120 **Thanks for the nice comment. The model COSMOS consists of the atmospheric module ECHAM5 and the ice-ocean module MPIOM, therefore, the ice-ocean model MPIOM used in our study is actually part of COSMOS. In this case, we are using the same GCM (COSMOS) and removing the atmosphere part to reduce the level of complexity (from**

coupled ECHAM5-MPIOM to ocean-only MPIOM). But in MPIOM it is not possible to separate the ocean from the sea ice. Therefore we choose another stand-alone sea ice model CICE.

125     So the only thing to be discussed here is the different components/forcings/configurations used in CICE and MPIOM. Therefore we add the following in the discussion section (line 429-436):

*It should be noted that CICE is in many aspects different from the sea ice component in MPIOM: 1) CICE uses the multi-layer approach with a sub-grid scale ice-thickness distribution (Bitz and Lipscomb, 1999), while MPIOM uses zero-layer thermodynamics following Semtner Jr (1976). 2) A submodel of ice dynamics based on an elastic-viscous-plastic rheology*

130 *(Hunke and Dukowicz, 1997; Hunke, 2001) is used in CICE, and in MPIOM viscous-plastic dynamics following Hibler III (1979) is applied. 3) Different spatial resolutions are used in CICE ($1°$) and MPIOM ($3°$). 4) CICE is forced by monthly-mean climatological data from National Center for Atmospheric Research (NCAR), while MPIOM experiments are forced by daily fields from the climatological OMIP data set (Röske, 2006). Therefore, the different model behavior between CICE and MPIOM can to a certain extent be explained by the different model configurations.*

135     i Section 2, l92-110: The discussion of the methods, the caveats related to each method, and how each method relate to previous studies should be streamlined. References should appear in parentheses for clarity and text that does not pertain to the Heat Flux Parameterisations should appear in other sections (e.g. Discussion).

    **Thanks. According to the comment, we have made several modifications to the manuscript:**

    **1). For the original l96-97, we put the references in a clearer way:**

140 *Laboratory experiments imply $35 \leq R \leq 70$ (Owen and Thomson, 1963; Yaglom and Kader, 1974; Notz et al., 2003).*

    **2). For the original l98-99, we remove the text into a new section of "Experimental design"**

    **3). For the original l104-111, we simplified the texts and also remove it to "Experimental design" section (189-195):**

*    For each model, we perform separate simulations based on the three ice-ocean heat flux formulations (see table 1). We assume that the 3-equation approach with $R = 35$ describes reality more realistically, and hence use this simulation as our*

145 *reference. In our idealized 1-D model, we also use $R = 70$ to test the model sensitivity with respect to this parameter. For a given value of $R$, we calculate $\alpha_h$ to satisfy the requirement described in McPhee et al. (1999). This results in a turbulent heat exchange coefficient $\alpha_h = 0.0095$ for $R = 35$ and $\alpha_h = 0.0135$ for $R = 70$. In SIM and CICE the mixed-layer salinity has a constant value of 34 g/kg. In MPIOM and COSMOS the salinity of the sea water evolves dynamically in response to oceanic or atmospheric processes.*

150     j l.206. The temperature at the interface for SIM-icebath is described here; yet it is not used in ice-bath. Are Tf and Tinterface from Equ. (2) and (3) for the ice-bath and 2-equation parameterizations (respectively) always the same? Does Tf for the ice-bath parameterization also depend on salinity? If so, that would also make it a "2 equations" problem. The differences between each" temperatures" in all three parameterizations should be described more clearly. A table with the formulas and temperatures used in each method would help clarify this issue.

155     **Yes, for ice-bath and 2-equation, the ice-ocean interface temperature is equal to the freezing temperature Tf. Tf in ice-bath also depend on salinity of sea water, but as the the sea water salinity in SIM and CICE is fixed at 34 psu, Tf is**

therefore always kept at -1.84 C. According to the comment, we now extend our table 1, in order to show the Tinterface and Tf in each experiment. We refer to Table 1 in our revised paper.

k l.99 Why is the sensitivity of the model to the parameter R tested given that "R=35 best describes reality"?

**Because laboratory experiments imply** $35 \leq R \leq 70$ **(Owen and Thomson, 1963; Yaglom and Kader, 1974; Notz et al., 2003). The value of** $R \approx 35$**, as suggested by McPhee et al. (2008) is more consistent with the laboratory result** ($35 \leq R \leq 70$)**, as compare to what Sirevaag (2009) found from an analysis of field data** ($R \approx 33$)**.**

l Section 4.1: This is where the authors examine the sensitivity of the choices described in Section 2 using an idealized model. A more detailed discussion of these results (along with additional figures) should be included in this section.

**We use our 1-D model to conduct a series of sensitivity studies to test the response of ice-ocean heat flux to the choices of mixed layer depth or friction velocity. We also add one figure here for the SIM model.**

**The following texts can also be found in the revised manuscript at line 239-248.**

*To quantify the different response of the simulated sea-ice cover for a larger range of forcing conditions, we carried out a series of sensitivity studies. For each of these, we varied one of the forcing parameters and analysed the difference in annual mean ice thickness between SIM-3eq35 and SIM-icebath.*

*We find that in our simplified setup, differences in ice thickness between SIM-3eq35 and SIM-icebath increases with mixed layer depth. This is due to the fact that the same amount of heat input from the atmosphere causes a smaller temperature change for a deeper mixed layer. According to equation (3), such smaller temperature change then causes a smaller heat flux to the ice bottom (Fig.2).*

*In addition, we find that the anomaly in sea ice thickness generally decreases with friction velocity. This is related to the fact that for larger friction velocities, more heat is transported to the ice-ocean interface in the 3-equation setup (Fig.2), which enhances sea-ice ablation. While in icebath assumption the ocean-to-ice heat flux is independent of the friction velocity.*

m Section 5: One question that keeps coming up in the readers mind while reading the results section is: did the model become more realistic as a result including a more realistic ice-ocean heat flux? Sometimes, especially in complex models, fixing one thing can often expose other problems, perhaps related to model tuning. With respect to this, the authors might consider describing how the results using the most realistic ice-ocean flux parameterization compare with observations. Sea ice thickness is tricky to measure, but Labe et al., 2018 could be a good start (They compare CESM, which uses CICE, to PIOMAS data). For the ocean surface, a qualitative comparison could be made to Peralta-Ferriz and Woodgate 2015, which is a nice study looking pan-Arctic ocean observations over the past 30 years.

**Thanks for the suggestion. We agree that a more realistic parameterization might not improve the model results. The results of COSMOS in terms of sea ice properties have been validated in Notz et al. (2013), which showed an overestimation of Arctic sea ice thickness in COSMOS (i.e., ECHAM5/MPIOM) as compared to the PIOMAS for both winter and summer. Therefore our implementation of the 3-equation approach increases such model bias. We now discuss this point in the last section of the updated paper (line 437-443).**

[Figure]

**Figure 1.** Anomalies of ocean-to-ice heat flux in SIM for (a) 3-equation minus icebath, and (b) 3-equation minus 2-equation, for different choices of mixed layer depth and friction velocity. Units are W/m$^2$.

*A detailed comparison of the simulations carried out here with observational data is beyond the scope of our study. However, we note that the sea ice thickness simulated by COSMOS have been evaluated by Notz et al. (2013), who found an overestimation of Arctic sea-ice thickness in ECHAM5/MPIOM (i.e., COSMOS) with the ice-bath formulation compared to the reanalysis from the Pan-Arctic Ice-Ocean Modeling and Assimilation System (PIOMAS) in both winter and summer. Improving the formulation of the ice–ocean heat flux by applying the 3-equation approach causes thicker ice and hence further increases this particular model bias. This indicates that the simplified heat flux parameterisation partly compensates for other errors in the coupled model setup.*

n l218 and lines below. l.218 should read "difference in annual mean ice thickness increases with friction velocity". The same form should be used for l220 and other instances that appear below: e.g. ". . .the larger the deeper our mixed layer is". The same comment for "mixed-layer and atmospheric temperature", and again for "ice concentration and ice thickness".Line 225: Do not word "explicable" should not be used in this context. Use instead: "This is explained by the fact that.."

**Thanks. We have corrected all the grammar errors mentioned here in the revised version.**

**3  Minor comments**

a l.9, abstract. 'The most realistic representation". This is vague. It should reference parameterization (1), (2) or (3) above.

**We now reference parameterization (1), (2) or (3) . (see line 13)**

b l.49-51. grammatical errors.

**We have correct it: "*This becomes particularly important when large amounts of meltwater accumulate underneath sea ice during summer.*"** (see line 74-75)

c l.53-57. This sentence should be broken into more than one sentence for clarity.

**We have separated the long sentence into several sentences (line 110-114):**

**"*Section 2 introduces various models that we use for our purposes: 1) A conceptual 1-dimensional model allows us to examine a wide parameter range. 2) The Los Alamos Sea Ice Model (CICE) allows us to determine changes of sea ice in a modern sea-ice model. 3) The Max-Planck-Institute Global Ocean/Sea-Ice Model (MPIOM) can be used for examining the impact of the parameterizations on the large-scale ocean circulation. 4) The fully coupled climate model COSMOS can further help us to look at the atmospheric response to the described parameterizations. *"**

d l.71.Tmix is defined as the ocean temperature. The vertical level (e.g. first layer) should be stated since the temperature is not constant in the mixed layer of CICE.

**CICE has only a slab-ocean, so there is only one fixed-depth mixed layer.**

e l.80. The freezing-point equation for seawater should be written explicitly.

**According to the comment, now we add a new equation describing the relationship between salinity and freezing point of sea water (Eq 4 at line 61 of the revised paper).**

f l.100. typo: salt and heat are transported almost equally efficiently.

**Thanks, we have corrected the typo in the updated version.**

g Section 3: There are several instances where numbers appear in the text or within equations without references. The references should be added.

**Thanks for the comment. For the bulk formulae we add the references for the Stefan-Boltzman constant and also for the thermal conductivity of sea ice. The equations for the idealized seasonal variation in various fluxes and surface albedo are based on approximation from observations. For this we provide the references in which the approximation had been made. We also provide the references for the observational seasonal variation in surface albedo, and monthly fluxes (we refer to section 2.1 in the revised paper).**

h l.121.Ts is the ICE surface temperature. This should be defined. Tbot should also be defined.

**Ts was defined right before the equation (line 125). Now we add the definition of Tbot in the updated manuscript (line 128).**

i l.116. The acronym MPIOM should be defined when it first appear.

**Now the acronym MPIOM is defined in the introduction:**

*" 3) The Max-Planck-Institute Global Ocean/Sea-Ice Model (MPIOM) can be used for examining the impact of the parameterizations on the large-scale ocean circulation."*

j l.127: A reference for ki should be given. Is this measurement really precise up to two decimal places?

**We re-run the SIM experiments with a new ki=2.03 (line 130-132).**

*For simplicity, we assume that the thermal conductivity of sea ice $k_i$ is constant and set $k_i = 2.03$ W/(m· K) according to the seminal 1-D thermodynamic sea-ice model of Maykut and Untersteiner (1971).*

k l.130 The Notz et al. and Maykut and Untersteiner references should appear above on line 129, i.e.before Fsw nd Fother is introduced.

**We have moved the references before the equations (line 136-137).**

l (5), l.130,l.131, l.135. The number in the equations should be replaced by symbols; the "×" should be removed and the numerical values should appear in the text, for the sake of clarity: (eg. 5.67×10-8 would be $\sigma$)

**According to the comment, we have improved our representation of equations (line 125-142):**

*The model consists of a simple zero-layer sea-ice model, where the surface temperature $T_s$ is determined by balancing atmospheric fluxes and the conductive heat flux through the ice according to*

$$-(1-\alpha)F_{sw} - F_{other} + \epsilon\sigma T_s^4 = -k_i \frac{T_s - T_{bot}}{h}. \tag{1}$$

*Here, $\alpha$ is the albedo of the ice surface, $F_{sw}$ is the short-wave flux, $\epsilon = 0.95$ the infrared emissivity, $\sigma = 5.67 \times 10^{-8}$ the Stefan-Boltzman constant, $T_{bot}$ the temperature at the ice-ocean interface, and $F_{other}$ is the sum of sensible heat flux,*

*latent heat flux and downward longwave radiation flux.* $(1 - \alpha)F_{sw} + F_{other}$ *represents the heat input to the surface of the ice, and* $\epsilon\sigma T_s^4$ *the upward longwave radiation flux from the ice surface. For simplicity, we assume that the thermal conductivity of sea ice* $k_i$ *is constant and set* $k_i = 2.03$ *W/(m· K) according to the 1-D thermodynamic sea-ice model of*

255 *Maykut and Untersteiner (1971). During melting periods, the surface temperature is fixed at the bulk freezing temperature of the ice and the excess heat is used to melt ice at the surface. We assume the sea ice in our idealized model to be very fresh, using a freezing temperature of 0 °C. At the ice bottom, the model calculates the change in ice thickness by balancing the conductive heat flux and the oceanic heat flux according to Eq. (??).*

*The seasonal variation of the atmospheric fluxes* $F_{sw}$ *and* $F_{other}$ *are prescribed according to the fits provided by Notz*

260 *(2005), approximating the monthly-mean observational data compiled by Maykut and Untersteiner (1971). These fits are:*

$$F_{sw} = A_1 exp[B(\frac{d - C_1}{D_1})^2] + E_1 \tag{2}$$

$$F_{other} = A_2 exp[B(\frac{d - C_2}{D_2})^2] + E_2 \tag{3}$$

*where* $A_1$*=19.5,* $A_2$*=117.8, B=-0.5,* $C_1$*=164.1,* $C_2$*=206,* $D_1$*=47.9,* $D_2$*=53.1,* $E_1$*=16.1,* $E_2$*=179.1, and* $d$ *is the number of the day in the year. The seasonal variation in surface albedo is calculated as*

$$\alpha = \frac{F}{1 + (\frac{d - G}{H})^2} + I \tag{4}$$

*where F, G, H and I have the values of -0.431, 207, 44.5 and 0.914, respectively. This equation is a fit to measurements*

265 *obtained during the Surface Heat Budget of the Arctic Ocean (SHEBA) campaign (Perovich et al., 1999).*

m l.126. The bulk freezing temperature of the ice should be defined.

**We assume the sea ice in our idealized model is very fresh (Si=0 psu) and thus have a freezing temperature of 0 degree. Now we have added this point in the text (line 133-134).**

n l.130. Numbers should be added for equations that appear after (5).

270 **We have labelled the three equations as (6), (7), and (8).**

o l.147: "the so-called CCSM3 set-up". This is not a commonly know term. The set-up should simply be defined. There are a lot of acronyms in the paper that are used and not defined. Acronyms should be defined when they first appear.

**Thanks. According to the comment, we now give full definitions for all the acronyms when they first appear in the paper, such as:**

275 **National Center for Atmospheric Research (NCAR)**

**Ocean Model Intercomparison Project (OMIP)**

**Surface Heat Budget of the Arctic Ocean (SHEBA)**

**The Los Alamos Sea Ice Model (CICE)**

280      p l.153 Is the mixed-layer salinity held constant in all the models? This would have an impact on the freezing temperature, and therefore on the heat flux parameterizations. This should be discussed (though not necessarily in the methods section).

     **In SIM and CICE the mixed-layer salinity is held constant (34 psu). But in MPIOM and COSMOS the salinity is treated as a passive tracer thus can be changed due to oceanic or atmospheric processes. Now we have clarified this point in the updated version.**

285      q l.154 What climatological dataset is being referred to exactly? The reference to the dataset should be included here. The time period over which the climatology was calculated should also be stated.

     **We now add in the paper (line 196-201):**

     *As atmospheric forcing, we use for our conceptual 1-D model equations (7) to (9), For CICE, we use the National Center for Atmospheric Research (NCAR) monthly-mean climatological data with $1° \times 1°$ resolution Kalnay et al. (1996). Input*
290 *fields contain monthly climatological sea-surface temperature, sea-surface salinity, the depth of ocean mixed layer, surface wind speeds, 10 m air temperature, humidity and radiation for the time period of 1984-2007. For MPIOM, we use a GR30 (about $3°$) horizontal resolution and 40 uneven vertical layers, forced by daily heat, freshwater and momentum fluxes as given by the climatological Ocean Model Intercomparison Project (OMIP) forcing (Röske, 2006).*

     r l.166: The ice distribution parameter must be described. The paper should be stand-alone.
295      **We now updated the texts (line 170-173):**

     *The change in sea ice thickness and concentration can be calculated via two main ice distribution parameters as outlined by Notz et al. (2013), the first one is a so-called lead closing parameter which describes how quickly the sea ice concentration increases during new ice formation processes, and the other describes the change in ice-thickness distribution during melting.*

300      s l.188, l.193: "We start with" is used twice.
     **We have rephrased the second sentence.**

     t l.191: "which is set to either 35 or 70". This is redundant as the simulations are listed above.
     **For simplifying the text we have removed those words from the manuscript.**

     u l.192: "no more changes from one year to the next." What were the initial changes?
305      **The initial ice thickness is 0.5 m in our idealized setup, which then changes with time due to the forcings applied. After several simulation years, the model finally ran into equilibrium. Since it is a simple 1-D model, there is no interannual variability in the final years. We found that after the simulation is in equilibrium, there are no changes from one year to the next.**

     v l.202: Why is the mixed-layer warming more slowly in SIM-2eq and 3eq? Is it that the ice cover is lost earlier when the
310 sun is higher over the horizon? This should be stated.

**Because in SIM-icebath the sea ice melts completely, so all the heat flux on ocean could directly lead to warming of sea water, while in in SIM-2eq and SIM-3eq, there are still remaining sea ice, so the heat flux is firstly used for melting of the sea ice.**

**Now we make this point clearer in the updated manuscript (line 224-228):**

*" Once the ice in SIM-icebath is melted completely, the ocean temperature rises rapidly and quickly exceeds that in SIM-2eq and SIM-3eq. This can be explained by two facts: 1) in SIM-2eq and SIM-3eq, the sea ice reflects most of the incoming shortwave radiation, and 2) in SIM-2eq and SIM-3eq the heat flux absorbed by open water is primarily used for sea-ice melting, while in SIM-icebath no more sea ice exists such that the entire heat flux into the ocean causes a warming of the sea water. "*

w l.208: The word "faster" should be used instead of "stronger" when describing "ice melt".

**We have changed the term to "faster".**

x Section 4.2: This subsection consists of a single paragraph that is approximately one page long. It should be broken into several paragraphs for clarity.

**Thanks for the comment. We have divided it into 5 paragraphs. We refer to section 4.2 in our revised paper.**

y l.240. The student's t-test should be more clearly described. Is it testing that the results from two different parameterizations are significantly different?

**Yes. We have clarified this in the revised paper (line 264-266).**

z Figures: Units and labels should be included on all figures.Fig.1. Units are missing.Fig.3-4. A larger font size for longitudes and the colorbars should be used.Fig.6-7. Units are missing for depth on the y axis.Fig.6-7. The seabed should appear in grey or a different color to differentiate it from zero in the water.Table 1: The parameter R should be defined/included in the Table caption. Currently, it is only defined later on l.192

**According to the comment, we have updated the figures in the revised manuscript.**

*To quantify the different response of the simulated sea-ice cover for a larger range of forcing conditions, we carried out a series of sensitivity studies. For each of these, we varied one of the forcing parameters and analysed the difference in annual mean ice-ocean heat flux between SIM-3eq35 and SIM-icebath.*

[Figure]

**Figure 1.** Anomalies of ocean-to-ice heat flux in SIM for (a) 3-equation minus icebath, and (b) 3-equation minus 2-equation, for different choices of mixed layer depth and friction velocity. Units are W/m$^2$.

*We find that in our simplified setup, differences in ice thickness between SIM-3eq35 and SIM-icebath increase with*
25  *mixed layer depth. This is due to the fact that the same amount of heat input causes a smaller temperature change for a*
*deeper mixed layer. According to equation 3, a smaller temperature change then leads to a smaller change in heat flux to*
*the ice bottom (Fig. 1).*

*In addition, we find that the anomaly in sea ice thickness generally decreases with friction velocity. This is related to*
*the fact that for larger friction velocity, more heat is transported to the ice-ocean interface in the 3-equation setup (Fig. 1),*
30  *which enhances sea-ice melt.*

2) I would like clarification on how the ice-bath model can be implemented in the ice-ocean coupled models.

**In ice-ocean models, in all grid cells covered by sea ice, one usually simply resets the temperature of the sea water in
the uppermost grid cell to its freezing point. All excess heat released during this adjustment is used to ablate sea ice.**

**Now we clarify it in more detail in the revised manuscript (line 173-178): "*In its standard setup, MPIOM uses an ice-***
35  ***bath parameterization to calculate the heat flux between the ocean and the ice. Wherever covered by sea ice, the temperature***
***of sea water in the uppermost grid cell is kept at its freezing point. All heat entering the uppermost grid cells either from***
***the atmosphere or from the deeper oceanic grid cell is instantaneously transported into the sea-ice cover, maintaining***
***the temperature of the uppermost oceanic layer at the freezing point. In the present study, this formulation is either used***
***directly, or replaced by the 2-equation or the 3-equation parameterization.*"**

40  3) Elaborate on the mechanisms that explain the weakening of the THC.

**The weakening of the AMOC in the ice-bath and 2eq as compared to 3eq35 is mainly affected by the NAO changes.
In the revised paper, we added a section to describe the relationship between NAO and AMOC (line 357-380):**

*In this section, the mechanism explaining the weakening AMOC in COSMOS-icebath and COSMOS-2eq as compared*
*to COSMOS-3eq35 is explored. It has long been recognized that the NAO variability has an important influence on the*
45  *AMOC (Curry et al., 1998; Delworth and Zeng, 2016). Variations in the NAO have been hypothesized to play a role in*
*AMOC variations by modifying air–sea fluxes of heat, water, and momentum. A similar relationship between NAO and the*
*AMOC has been reported also for past climate conditions (Shi and Lohmann, 2016; Shi et al., 2020). Here in Fig.11we show*
*the results from a composite analysis between the NAO index and the anomalies in mixed layer depth based on COSMOS-*
*3eq35. It is calculated by averaging March mixed layer depth anomalies (departure from the mean state) during years when*
50  *the NAO index exceeds one standard deviation.*

*The convective activities in the Labrador Sea and the Greenland-Iceland-Norwegian (GIN) Seas are shown to have*
*important contributions to the production and transport of North Atlantic deep water (Fig.11a). For comparison we also*
*show the distribution of mixed layer depth in MPIOM-3eq35 in Fig. S3, which indicates a different location of main deep*
*water convection site in our ice-ocean coupled model: the northeastern North Atlantic. The results from the composite*
55  *analysis shown in Fig.11b indicate that the anomalous NAO pattern can lead to significant changes in the ocean circulation.*
*We find that the intensity of the Labrador Sea convection is characterized by variations that appear to be synchronized*

*with variabilities in the NAO. Therefore, the weakening of AMOC in our simplified setups compared to the most realistic approach can be attributed to the simulated anomalous negative NAO phase.*

*The NAO affects the sea water convection mainly via modifying the surface heat fluxes which leads to anomalies in the* spatial and vertical density gradient. Fig.12a shows the composite map between surface heat flux anomalies and the NAO index. During the positive phase of NAO, more heat than usual is removed from the ocean to the atmosphere at the western Atlantic, in particular the Labrador Sea. Such pattern is in good agreement with the NAO-relative heat flux anomalies derived from the European Centre for Medium-Range Weather Forecasts (ECMWF) interim reanalysis (Delworth and Zeng, 2016). The enhanced removal of heat favors an increase in the surface density and thereby strengthens deep water formation. On the other hand, the NAO also affects the net precipitation over North Atlantic Ocean. As illustrated in Fig.12b, relatively drier condition could occur over Labrador Sea and Irminger Sea during positive NAO years. .*

4) How does the mixed layer look like in the ice-ocean coupled models (i.e. depth,...)

**Following this comment, we now show the mixed layer depth of MPIOM in the Supplementary Fig. S3.**

**In the revised paper we added (line 365-368):**

*" The convective activities in the Labrador Sea and the Greenland-Iceland-Norwegian (GIN) Seas are shown to have important contributions to the production and transport of North Atlantic deep water (Fig. ??a). For comparison we also show the distribution of mixed layer depth in MPIOM-3eq35 in Fig. S3, which indicates a different location of main deep water convection site in our ice-ocean coupled model: the northeastern North Atlantic. "*

5) Scales on figures are chosen to essentially show the sign but not so much the magnitude of the differences (i.e, one can tell where ice is thicker or thinner but not by how much). Is that to hide the large differences that cannot be easily explained between model setups?

**The blue-red color-bar was chosen as it represent clearly the sign of the changes. In the revised version we have chosen a different color-bar so the magnitude of the anomalies can be more easily read. Here we refer to the figures in the updated manuscript.**

6) If the mixed layer temperature is so critical in controlling the temperature of the deep waters then it is all the more important to give a convincing description of its evolution and realism

**Following the reviewer's comment, we now plot the anomalies of mixed layer temperature, as seen in Fig. 2 and Fig. 3 in the present response letter. We find the patterns are very similar to the SST anomalies as shown in the manuscript. As the sea water in the mixed layer is well mixed, it is not surprising that the mixed layer temperature anomalies mimic that of the SST.**

7) A comparison with Tsamados et al (2015) would be useful especially as the author of this study found a small impact of the 3eq BC. Discuss

**Thanks for the comment, now we mention the study by Tsamados et al (2015) in the introduction and discuss it in the discussion section:**

**In introduction (line 92-105):**

[Figure]

**Figure 2.** Anomalies of mixed layer temperature for (a) MPIOM-3eq35 minus MPIOM-icebath in March, (b) MPIOM-3eq35 minus MPIOM-2eq in March, (c) MPIOM-3eq35 minus MPIOM-icebath in September, and (d) MPIOM-3eq35 minus MPIOM-2eq in September. Units: K.

*Exploring the behavior of different parameterizations describing ice-ocean heat flux has been an important topic in model studies. Significant differences can generally exist between melt rates calculated with the 3-equation approach and less realistic approaches (c.f. Notz et al., 2003; Tsamados et al., 2015), as only the 3-equation approach allows for heat fluxes that are directed from the interface into the water and therefore allows for a realistic limitation of melt rates through the*
95  *formation of a fresh water layer underneath the ice. In a recent study the sensitivity of sea ice simulation to the approaches*

[Figure]

**Figure 3.** As in Fig. 2 but for COSMOS.

*introduced in McPhee (1992) and Notz et al. (2003) have been examined using a stand-alone sea ice model CICE (Tsamados et al., 2015). CICE uses a simple thermodynamic slab ocean with fixed mixed layer depth and sea water salinity. Thus, the realistic effect of oceanic processes can not be represented. For example the sea ice over the Southern Ocean is severely overestimated by CICE due to a lack of warming effect from the Antarctic deep water. Therefore it is necessary to also*

100 *investigate the ice-ocean heat flux formulations in a more complex system, including an interactive ocean or even the atmosphere. Based on this motivation, in the present study, we examine how different physical realism that is represented by the three discussed parameterizations impact the resulting ice cover, large-scale oceanic circulation, and atmosphere*

*properties in different numerical models ranging from an idealized columnar model, a stand-alone sea ice model, an ice-ocean coupled model, to a most complex climate system model.*

**In the discussion we add (line 400-406):**

*Note that our results are in good agreement with a previous study using CICE (Tsamados et al., 2015) that found in August stronger basal -melting of Arctic sea ice, decreased Arctic Ocean salinity, cooling of sea water in the central Arctic and warming of sea water at the ice edge in the 2-equation experiments compared to the 3-equation approach. However, in their study the effects are more pronounced, possibly because we used different model versions of CICE and different parameters for the ice-ocean heat flux formulations, one example is the value for R, which is 50 in Tsamados et al. (2015) and 35 in our case. In addition, different atmospheric forcings were used in the two studies.*

8) Is the most advanced thicker because of the reduced (even reversed) summer fluxes from the ocean to the ice? Again why is this results not so marked in Tsamados et al(2015)

**Thanks for the comment. In fig.6 and fig.9 of Tsamados et al(2015), we can see that our results regarding the bottom melt and thickness of sea ice are identical. Both our studies indicate reduced sea ice when using the 2-eq rather than the 3-eq approach. This is due to a larger bottom melt rate in the 2-eq experiment. From fig.9 of Tsamados et al(2015), we see that the anomaly of summer sea ice thickness in 2-eq as compared to 3-eq is up to -0.25 m, while in our study the maximum change is -0.1 m (Fig. 2d in the manuscript). Therefore actually Tsamados et al(2015) shows a stronger change than our study does. Another thing to be noted is that Tsamados et al(2015) and our study use different values for the exchange coefficients for salinity and heat, which is R=50 in Tsamados et al(2015), while we have R=35. Now we discuss this point in the discussion (line 400-406):**

*Note that our results are in good agreement with a previous study using CICE (Tsamados et al., 2015) that found in August stronger basal -melting of Arctic sea ice, decreased Arctic Ocean salinity, cooling of sea water in the central Arctic and warming of sea water at the ice edge in the 2-equation experiments compared to the 3-equation approach. However, in their study the effects are more pronounced, possibly because we used different model versions of CICE and different parameters for the ice-ocean heat flux formulations, one example is the value for R, which is 50 in Tsamados et al. (2015) and 35 in our case. In addition, different atmospheric forcings were used in our studies.*

9) I am really uncertain as to the significance of the impacts found on the ocean and atmosphere. How does this compare to internal variability within the models? I have heard in the past Notz state that sea ice physics does not play a significant role in the climate response (I might be misquoting and apologies if I do) but how do these finding square with this view? 10) Are the results presented reproducible. What if you analysed another 10 years or 100 years period?

**Thanks, here we answer comments 9 and 10 together, as they both point to the significance of our results. We agree that showing the robustness of the results is crucial. To examine the significance of our results, we have performed a Student t-test which takes the internal variability into consideration. In the anomaly plots, we marked the areas with significant level higher than 95 percent with black dots. As seen from those figures, our main results are beyond the internal variability of the climate. For the sea level pressure pattern, no statistical significant changes are found, as the**

SLP variability is very large. Therefore we analyzed another 100 years of the simulation, and found consistent pattern as in the last 100 model years. These give us the confidence in the robustness of our results. (Note from Dirk Notz: I often say that in my view, in current models one primarily needs to improve the atmospheric/oceanic forcing in order to improve the simulation of the sea ice cover. In the opposite direction, things are less clear cut as indicated by the debate on the impact of sea-ice loss on Mid-Latitude weather patterns. However, also here in my view the impact of sea ice on atmospheric processes is small compared to internal variability, consistent with what we find here. For the ocean, I don't think I ever intended to say that the role of sea ice is negligible or the like, primarily because of its impact on watermass-transformation.)

11) I wonder why you do not use the same setup to analyse several other sea ice processes (albedo, melt pond, form drag, as per the recipe of Tsamados et al 2015 etc....). Is it too good to be true?

We agree with the reviewer that there are many other interesting processes/parameters that deserve being examined. However, the focus of this study is the impacts of different parameterizations of ice-ocean heat exchange on simulated ice thickness, ice concentration, and water masses in a hierarchy of models, 1D sea ice model, basin-scale stand-alone sea ice model, ice-ocean coupled model, and fully coupled model.

Our work started in the year 2011, therefore most of the simulations were performed before the publication of Tsamados et al (2015). We used a super long time to put everything together into a complete paper. The initial idea of our work is based on my interest on the 3-eq heat flux parameterization as described in Notz et al (2003), that is why I came to Germany and work with Prof. Notz on the topic. We are also interested in applying various approaches describing various processes (albedo effect, lateral melting, top melting, form drag...) into various models. This could be our next step.

12) Not clear how the prescribed atmospheric forcing subdues the impact on AMOC. Please elaborate.

The AMOC consists of wind-driven circulation (WDC) and thermohaline circulation (THC). The atmospheric process over North Atlantic plays an important role on determining the AMOC state. One of the key elements controlling the atmospheric circulation over the North Atlantic is NAO. As discussed in our paper (as well as many other papers), the positive phase of NAO leads to stronger deep mixing at the North Atlantic subpolar regions. But in ice-ocean model, as the atmosphere are prescribed, there is no difference in the atmospheric state between different experiments. Therefore, the change of AMOC in the MPIOM simulations can only be affected by thermohaline state, while in COSMOS runs both atmospheric and thermohaline conditions play a role.

We have discussed this point in the discussion section (line 416-420).

*"As indicated in the present paper and many other studies (Curry et al., 1998; Latif et al., 2006; Sun et al., 2015), thee AMOC is closely related to the atmospheric processes over North Atlantic Ocean. One of the key elements controlling the atmospheric circulation over the North Atlantic is the NAO. As the atmospheric forcings are prescribed in MPIOM, there is no difference in the atmospheric state among the MPIOM experiments. Therefore, the prescribed atmospheric forcing largely limits the air-sea interaction feedback."*

**2 Specific comments:**

P1 L18 expand on motivation and justification

**We now expand our motivation and justification in the end of introduction (line 92-105):**

*Exploring the behavior of different parameterizations describing ice-ocean heat flux has been an important topic in*

175 *model studies. Significant differences can generally exist between melt rates calculated with the 3-equation approach and less realistic approaches (c.f. Notz et al., 2003; Tsamados et al., 2015), as only the 3-equation approach allows for heat fluxes that are directed from the interface into the water and therefore allows for a realistic limitation of melt rates through the formation of a fresh water layer underneath the ice. In a recent study the sensitivity of sea ice simulation to the approaches introduced in McPhee (1992) and Notz et al. (2003) have been examined using a stand-alone sea ice model CICE (Tsamados*

180 *et al., 2015). CICE uses a simple thermodynamic slab ocean with fixed mixed layer depth and sea water salinity. Thus, the realistic effect of oceanic processes can not be represented. For example the sea ice over the Southern Ocean is severely overestimated by CICE due to a lack of warming effect from the Antarctic deep water. Therefore it is necessary to also investigate the ice-ocean heat flux formulations in a more complex system, including an interactive ocean or even the atmosphere. Based on this motivation, in the present study, we examine how different physical realism that is represented*

185 *by the three discussed parameterizations impact the resulting ice cover, large-scale oceanic circulation, and atmosphere properties in different numerical models ranging from an idealized columnar model, a stand-alone sea ice model, an ice-ocean coupled model, to a most complex climate system model.*

P1 L20 clarify this paragraph. Which freezing temperature...

**We mean the freezing point of the uppermost cell itself. Here we improved the texts:**

190 *The temperature of the uppermost ocean grid cells is fixed at its freezing temperature.*

P1 L27 Here is a good place to expand on the analogy and differences between momentum and heat transfer and write the equations and if need be criticise what is missing in either of them. At the moment it is too vague. For example what do you 'differs from the exchange of momentum" in what way?

**At line 42-47:**

195 *These measurements demonstrate that in particular during melting, the exchange of scalar quantities such as heat and salt differs significantly from the exchange of momentum. The reason for this lies in the fact that, unlike ice-ocean momentum flux, heat and mass transfer are strongly affected by a thin sublayer controlled by molecular processes (McPhee et al., 1987). Consistent with laboratory studies of heat transfer over hydraulically rough surfaces (Yaglom and Kader, 1974), the heat exchange is hence not only determined by turbulent processes but also by diffusion through this molecular sublayer.*

200 P2 L42 a local phase

**Thanks for the correction. We have modified the text.**

P2 L50 , Tsamados et al (2015)

**We have added the reference.**

P4 L85 together with the freezing equation(1)

205 **We have modified the text accordingly.**

P4 L 91 2003) and as implemented in CICE by Tsamados et al (2015)

**Thanks for providing the reference, we are glad to cite Tsamados et al (2015) in our revised paper.**

P5 L137 fixed mixed layer depth?

**Yes. Our SIM is a simple 1-D sea ice model with a fixed-depth mixed layer.**

210 P6 L141 version? Expand + maybe

**We used the 4.0 version, we now clarify this point in the paper (line 146-151):**

*We use the 4.0 version of the stand-alone sea ice model CICE to investigate the sensitivity of sea ice to the three ice-ocean heat flux parameterizations in a modern sea-ice model. The model consists of a multi-layer energy-conserving thermodynamic sub-model (Bitz and Lipscomb, 1999) with a sub-grid scale ice-thickness distribution, and a submodel of ice dynamics*

215 *based on an elastic-viscous-plastic rheology (Hunke and Dukowicz, 1997; Hunke, 2001) that uses incremental remapping for ice advection (Lipscomb and Hunke, 2004). A detailed model description is given in Hunke and Lipscomb (2010).*

P6 L150 again default mixed layer of fixed depth. Not realistic, this affects your Tmix and hence your results. P6L153 same issue with salinity should change with mixed layer depth

**We agree, in an ice-ocean model like the MPIOM the salinity changes with ocean depth, and the depth of the mixed**
220 **layer is not fixed. What is described here is the stand-alone sea ice model CICE. It has only one ocean layer (a slab-ocean), and in its standard setup, 34 psu is used for the ocean salinity. That is one of the reasons why we also tested the 3 parameterizations in more complex models such as MPIOM and COSMOS. This is also a highlighting point of our study, as most previous studies on testing sea ice parameterizations were only based on stand-alone sea ice models.**

**We talked about this point in the introduction section (line 97-101):** *"CICE uses a simple thermodynamic slab ocean*
225 *with fixed mixed layer depth and sea water salinity. Thus, the realistic effect of oceanic processes can not be represented. For example the sea ice over the Southern Ocean is severely overestimated by CICE due to a lack of warming effect from the Antarctic deep water. Therefore it is necessary to also investigate the ice-ocean heat flux formulations in a more complex system, including an interactive ocean or even the atmosphere."*

P6 L160 expand

230 **Thanks for the suggestion, we now expand the description for MPIOM (line 162-165):**

*MPIOM is based on the primitive equations with representation of thermodynamic processes (Marsland et al., 2003). The orthogonal curvilinear grid is applied in MPIOM with the north pole located over Greenland. The relevant terms of the surface heat balance are parameterized according to bulk formulae for turbulent fluxes (Oberhuber, 1993), and radiant fluxes (Berliand, 1952).*

235 P6 L165 The repartition? Rephrase slightly

**We improved the text to let it be more understandable (line 170-173):**

*The change in sea ice thickness and concentration can be calculated via two main ice distribution parameters as outlined by Notz et al. (2013), the first one is a so-called lead closing parameter which describes how quickly the sea ice concentration increases during new ice formation processes, and the other describes the change in ice-thickness distribution during melting. "*

P6 L169 wouldn't it better to have it at 1 deg and run for 100 years? Or is that needed for equilibration?

**For ice-ocean model, a equilibrium-state requires a stable AMOC, which normally needs a simulation time of several centuries. That's why we run both MPIOM and COSMOS for 1,000 years.**

**Regarding the spatial resolution, we choose GR30 to be consistent with COSMOS, as in COSMOS the standard setup of spatial resolution is GR30. This also helps us to save computing resources.**

P8 L187 and ice concentration P8 L210 more slowly P10 L221 a smaller...for a deeper... P10 L226 are larger...

**Thanks, we have corrected the above mentioned grammar errors.**

P10 L238 I am surprised that you needed a 90 years spin-up for a stand-alone sea ice model.

**We let all models run at least 100 simulation years, although the stand-alone sea ice model reach the equilibrium in a few years of integration. For more complex models we run the simulations for 1,000 years.**

P13 L257 what do you mean by far-field? Mixed layer? Also typo...at the interface

**Sorry for the confusion. We mean "the uppermost ocean layer". We have corrected this as well as the typo.**

P13 L268 I am not clear on how you can obtain an ice bath situation in the ice-ocean coupled models

**MPIOM and COSMOS originally uses the ice-bath assumption.**

**Now we clarify it in more detail in the revised manuscript (line 173-177): "** *In its standard setup, MPIOM uses an ice-bath parameterization to calculate the heat flux between the ocean and the ice. Wherever covered by sea ice, the temperature of sea water at the uppermost grid cell is kept at its freezing point. All heat, coming from above atmosphere or deeper ocean, will then be instantaneously transported to sea ice, thus keeping the first-layer temperature unchanged. In the present study, this formulation is either used directly or replaced by the 2-equation or the 3-equation parameterization.***"**

P13 L269 how significant is this cooling in COSMOS? How does it compare to model internal variability for example?

**To test the significance level of the anomalies, we performed Student t-test. In the revised plots, the areas with significant changes (significance level > 95%) are marked with dots. As shown in figure 4i-k, the cooling of the North Atlantic in COSMOS is significant (marked with black dots), and thus the anomalies are beyond the internal variability. However, in Fig.4l, the cooling over North Atlantic in September between 2eq and 3eq35 is not statistically significant.**

Fig5why don't you show CICE?

**CICE is a stand-alone sea ice model which includes only a slab ocean mixed-layer parameterization, but the salinity of the ocean is kept constantly at 34 psu. That is why it is not necessary to show the salinity change for CICE. The salinity anomaly should be 0 everywhere in CICE.**

P13 L274 doesn't less ice mean more growth in winter(negative feedback) and hence more brine release?

270   **According to the reviewer's comment, I think we shall use "larger melting rate" rather than "reduction in sea ice thickness" here, as the salinity of the sea water responds to the change of sea ice, instead of the absolute value of sea ice. Now we have corrected this point in the manusscript (line 298-301).**

*"In regions in which the ice-bath approach or the 2-equation approach cause an increased heat flux to the ice underside, and hence a larger melting rate of sea ice in summer as well as a smaller growth rate in winter, the ocean is generally less*
275   *salty in these simulations with a simplified parameterization of ice–ocean heat exchange than in the simulations with the full 3-equation parameterization."*

P13 L280 the small differences between COSMOS-2eq and COSMOS-3eq35 indicates that once mixed layer allowed to evolve impact of this parameterisation is small?

**Regarding the small differences in COSMOS, we have discussed in the discussion section (line 421-428):**

280   *"Our study indicates a less pronounced sea-ice response to ice-ocean interface heat flux parameterizations in the fully coupled climate model COSMOS than in the ice-ocean model MPIOM (compare Fig.3e-h with Fig.3i-l). This is because the change of the AMOC has a dampening effect on the simulated sea ice anomalies. The strengthening of the AMOC in COSMOS-3eq can lead to a warming over the Northern Hemisphere, especially over the North Atlantic and the Arctic. This hypothesized link between the AMOC and Northern Hemisphere mean surface climate has been documented in an*
285   *abundance of studies (e.g., Schlesinger and Ramankutty, 1994; Rühlemann et al., 2004; Dima and Lohmann, 2007; Parker et al., 2007). The AMOC-induced warming helps to reduce the sea ice mass over the Arctic and North Atlantic subpolar regions. Indeed, the sea ice across the Greenland Sea and Baffin Bay are found to be thinnest in COSMOS-3eq."*

P13 L278 I don't get this explanation

Interestingly, the opposite sign is observed in the Barents sea and its adjacent regions (Fig.6e-f), despite the larger melt rates
290   in the ice-bath scheme. This is likely due to sea ice dynamic processes. The interannual variability of sea ice volume in the Barents sea and its adjacent regions is mainly determined by variations in sea ice import from the Central Arctic, rather than the local thermodynamic change of sea ice (Koenigk et al., 2009). Since less Arctic sea ice is simulated in COSMOS-icebath, less sea ice volume is transported to the subpolar regions in the experiment with ice-bath parameterization. However, it is just a speculation. The sea ice dynamic process in our simulations needs further investigation. Therefore in our revised paper, we
295   removed this part and we will have a more detailed analysis on the sea ice dynamics in the future.

P17 L288 where are these regions of deep water formation?

**Thanks, we mean the marginal ice zone. We now corrected the texts (line 311-313):**

*"This warming in the simulations with the least realistic parameterization of ice–ocean heat exchange reflects its earlier ice loss in in the marginal ice zone."*

300   P17 L289 I am not sure I follow why thinner means fresher mixed layer. The system rapidly equilibrates to a thinner state and then no reason to have fresher ML

**Thanks for the comment. We agree on that. At the same time, the system also equilibrates to a state with more summer ablation of sea ice. This effect dominates the change in salinity. So it is the change in the melting rate rather**

**than the sea ice thickness that leads to sea water freshening. Now we have corrected this in the updated manuscript**

305 **(line 314-316):**

*"As the simplified parameterizations both lead to faster melting rate of sea ice at the Arctic Ocean in summer, and less growing rate of sea ice in winter, as compared to the most realistic approach, one would expect a freshening of the ocean mixed layer and the deep water mass that originates from such fresher surface source water."*

P17 L293 could it be because these results are coincidental

310 **The anomalies are statistically significant as they pass 95% significance level based on Student's t-test. On the other hand, the anomaly patterns from ice-bath and 2-eq experiments are identical, only with different magnitudes.**

Fig S2 caption -> departure from

**Thanks, we have corrected the typo in the revised version.**

**References**

[revised manuscript text omitted]

---

## Referee Report (RR1)

Author: Sensitivity of Northern Hemisphere climate to ice-ocean interface heat flux parameterizations

Authors: Xiaoxu Shi, Dirk Notz, Jiping Liu, Hu Yang, and Gerrit Lohmann

This is a second review of the paper by Shi et al. The authors have addressed the comments of the reviewers satisfactorily. I recommend that the paper be accepted after the minor revisions below have been addressed.

Minor points:

1- The beginning of the introduction with the presentation of an equation in the second sentence is a little brutal. I would suggest adding a paragraph introducing the context of the work; the preferred approach (ice bath, 2 or 3-equations model) in the GCM community; etc.
2- Line 33: "… at its freezing POINT temperature…".
3- Line 41: Eliminate double parentheses before MIZEX.
4- Line 96-97: "In a recent study…" This sentence must be edited. Punctuation should be used given the large number of subordinate propositions that preface the principal proposition, or better still, it should be written in the active form. This writing style is used throughout. I suggest doing this everywhere necessary everywhere necessary. Line 408 is a prime example of this.
5- Line 109: Have you included your 3-equation model into the public version of CICE? CICE is on github and publicly available. It would a nice addition/contribution to the community (contacts: JF Lemieux, ECCC and E. Hunke, LANL).
6- Line 312: "… reflects is caused…" kill one of the two.
7- Figure 7 caption: State the latitude of the transect.

Bruno Tremblay, McGill University.

---

## Author Response (AR2)

**Response letter to Reviewer 1**

Xiaoxu Shi[1], Dirk Notz[2,3], Jiping Liu[4], Hu Yang[1], and Gerrit Lohmann[1]

[1]Alfred Wegener Institute, Helmholtz center for Polar and Marine Research, Bremerhaven, Germany
[2]Institute for Oceanography, Center for Earth System Research and Sustainability (CEN), Hamburg University, Hamburg, Germany
[3]Max Planck Institute for Meteorology, Hamburg, Germany
[4]Department of Atmospheric and Environmental Sciences, University at Albany, New York, USA

**Correspondence:** Xiaoxu Shi (xshi@awi.de)

Review of Sensitivity of Northern Hemisphere climate to ice-ocean interface heat flux parameterizations by Shi et al

This is a second review of thepaper by Shi et al. The authors have addressed the comments of the reviewers satisfactorily. I recommend that the paper be accepted after the minor revisions below have been addressed.

**Dear Reviewer,**

5 **Thanks so much for your positive feedback. In the following we present our point-to-point responses. Our answers to your comments are written in bold.**

**Thanks again for your time and efforts.**

**Best, Xiaoxu**

**1 Minor points:**

10 1-The beginning of the introduction with the presentation of an equation in the second sentence is a little brutal. I would suggest adding a paragraph introducing the context of the work; the preferred approach (ice bath, 2 or 3-equations model) in the GCM community; etc.

**We agree with the reviewer. To make the structure of the paper better and more clear, in the revised version we have made a new section describing in detail each of the three parameterizations. We refer to Section 2 of the updated paper.**

15 2-Line 33: "... at its freezing POINT temperature..."

**We have corrected it based on the comment.**

3-Line 41: Eliminate double parentheses before MIZEX.

**We changed the texts into:**

*"However, the ice-bath paradigm is incompatible with observations, i.e., the 1984 Marginal Ice Zone Experiment (MIZEX),*
20 *which clearly indicates that an ice-covered mixed layer can store significant amounts of heat."*

4-Line 96-97: "In a recent study..." This sentence must be edited. Punctuation should be used given the large number of subordinate propositionsthat preface the principal proposition, or better still, it should be written in the active form.This writing style is used throughout. I suggest doing this everywhere necessary everywhere necessary. Line 408 is a prime example of this.

**Following this comment, we now re-wrote the sentence using active form:**

*"A previous study has examined the sensitivity of sea ice simulation to the approaches introduced in ? and ? using a stand-alone sea ice model CICE "*

5-Line 109: Have you included your 3-equation model into the public version of CICE? CICE is on github and publicly available. It would a nice addition/contribution to the community(contacts: JF Lemieux, ECCC and E. Hunke, LANL).

**Our source codes are available online, here we refer to our section of "Code and data availability. Our next step is to implement our code into the latest release version of CICE6.**

6-Line 312: "... reflects is caused..." kill one of the two.

**Following the reviewer's comment, we now re-phrase the texts into:**

*"This warming in the simulations with the least realistic parameterization of ice–ocean heat exchange reflects the earlier ice loss in the marginal ice zone, which causes enhanced surface warming of the water there."*

7-Figure 7 caption: State the latitude of the transect.

**We now describe in the caption:**

*"Anomalies in zonal mean temperature and salinity vertical profile across the North Atlantic section (-80-0°W) for the latitudes from 30°S to 90°N."*

**Sensitivity of Northern Hemisphere climate to ice-ocean interface heat flux parameterizations**

Xiaoxu Shi[1], Dirk Notz[2,3], Jiping Liu[4], Hu Yang[1], and Gerrit Lohmann[1]

[1]Alfred Wegener Institute, Helmholtz center for Polar and Marine Research, Bremerhaven, Germany
[2]Institute for Oceanography, Center for Earth System Research and Sustainability (CEN), Hamburg University, Hamburg, Germany
[3]Max Planck Institute for Meteorology, Hamburg, Germany
[4]Department of Atmospheric and Environmental Sciences, University at Albany, New York, USA

**Correspondence:** Xiaoxu Shi (xshi@awi.de)

**Abstract.** We investigate the impact of three different parameterizations of ice-ocean heat exchange on modeled sea-ice thickness, sea-ice concentration, and water masses. These three parameterizations are (1) an ice-bath assumption with the ocean temperature fixed at the freezing temperature; (2) a 2-equation turbulent heat-flux parameterization with ice-ocean heat exchange depending linearly on the temperature difference between the underlying ocean and the ice-ocean interface whose temperature is kept at the freezing point of the sea water; and (3) a 3-equation turbulent heat-flux approach in which the ice-ocean heat flux depends on the temperature difference between the underlying ocean and the ice-ocean interface whose temperature is calculated based on the local salinity set by the ice-ablation rate. Based on model simulations with the standalone sea-ice model CICE, the ice-ocean model MPIOM and the climate model COSMOS, we find that compared to the most complex parameterization (3), the approaches (1) and (2) result in thinner Arctic sea ice, cooler water beneath high-concentration ice and warmer water towards the ice edge, and a lower salinity in the Arctic Ocean mixed layer. In particular, parameterisation (1) results in the smallest sea ice thickness among the 3 parameterizations, as in this parameterisation all potential heat in the underlying ocean is used for the melting of the sea ice above. For the same reason, the upper ocean layer of the central Arctic is cooler when using parameterisation (1) compared to (2) and (3). Finally, in the fully coupled climate model COSMOS, parameterisations (1) and (2) result in a fairly similar oceanic or atmospheric circulation. In contrast, the most realistic parameterization (3) leads to an enhanced Atlantic meridional overturning circulation (AMOC), a more positive North Atlantic Oscillation (NAO) mode and a weakened Aleutian Low.

*Copyright statement.* This study presents original work, and it has not been submitted elsewhere before. All authors agree to the submission.

[revised manuscript text omitted]